# Stress-driven potentiation of lateral hypothalamic synapses onto ventral tegmental area dopamine neurons causes increased consumption of palatable food

Louisa E. Linders [1], Lefkothea Patrikiou[1], Mariano Soiza-Reilly [2], Evelien H. S. Schut[1], Bram F. van Schaffelaar[1], Leonard Böger[1], Inge G. Wolterink-Donselaar[1], Mieneke C. M. Luijendijk[1], Roger A. H. Adan [1] & Frank J. Meye [1] ✉

Stress can cause overconsumption of palatable high caloric food. Despite the important role of stress eating in obesity and (binge) eating disorders, its underlying neural mechanisms remain unclear. Here we demonstrate in mice that stress alters lateral hypothalamic area (LHA) control over the ventral tegmental area (VTA), thereby promoting overconsumption of palatable food. Specifically, we show that glutamatergic LHA neurons projecting to the VTA are activated by social stress, after which their synapses onto dopamine neurons are potentiated via AMPA receptor subunit alterations. We find that stress-driven strengthening of these specific synapses increases LHA control over dopamine output in key target areas like the prefrontal cortex. Finally, we demonstrate that while inducing LHA-VTA glutamatergic potentiation increases palatable fat intake, reducing stress-driven potentiation of this connection prevents such stress eating. Overall, this study provides insights in the neural circuit adaptations caused by stress that drive overconsumption of palatable food.

Stress can increase the intake of rewarding calorie-dense food (e.g. fat and sugar)[1,2], posing problems for people prone to develop obesity or (binge) eating disorders[3,4]. The neural basis of such stress eating remains largely unclear, but likely involves alterations in the reward system[3,5]. The ventral tegmental area (VTA), a midbrain structure comprising a large population of dopaminergic (VTA$_{DA}$) neurons, is a potential candidate hub in encoding stress eating. VTA$_{DA}$ neurons play a critical role in food reward processing and motivation[5–7]. Moreover, VTA$_{DA}$ neurons and their output targets like the prefrontal cortex are sensitive to stressful stimuli[8–11]. Finally, repeated social stress in mice increases preference for palatable fat, and this effect requires catecholaminergic neurons (which VTA$_{DA}$ neurons are part of)[12]. However,

it remains unknown whether and how stress alters VTA$_{DA}$ circuitry to promote the intake of palatable food.

Previous studies in rodents have reported that stressful events strengthen glutamatergic synapses onto VTA$_{DA}$ neurons[13–17], but the origin of these synaptic inputs remains unknown. The lateral hypothalamic area (LHA), which sends glutamatergic and GABAergic projections to multiple VTA cell types[18–23], is a likely candidate in this regard. Indeed, the LHA has been extensively implicated in regulating (palatable) food intake[20,21,23,24]. Stimulation of LHA GABA neurons has previously been reported to directly increase food intake[20,21,23]. Stimulation of LHA glutamatergic (LHA$_{glu}$) projections to the VTA does not immediately affect feeding in naïve non-stressed conditions[20,21].

[1]Department of Translational Neuroscience, Brain Center, UMC Utrecht, Utrecht University, Utrecht, The Netherlands. [2]Instituto de Fisiología, Biología Molecular y Neurociencias (IFIBYNE), CONICET, University of Buenos Aires, Buenos Aires, Argentina. ✉e-mail: F.J.Meye-2@umcutrecht.nl

However, LHA$_{glu}$ neurons projecting to the VTA exhibit calcium reactivity during food intake and their role in feeding may be dependent on internal states[25]. Moreover, LHA$_{glu}$ projections to the VTA are sensitive to stressful stimuli[22,26]. Despite these insights, it remains unknown if stress causes LHA-VTA synaptic plasticity, and whether this drives palatable food choices. Here we test this hypothesis using a social stress model that recapitulates stress eating responses. We combine this with ex vivo neural circuit plasticity unravelling approaches that then guide manipulations of LHA-VTA stress plasticity in vivo. We demonstrate that stress-driven LHA$_{glu}$-VTA$_{DA}$ potentiation is both necessary and sufficient for enhancing palatable food intake post stress.

## Results
### Social stress increases palatable food intake and preference
We utilized a social stress paradigm to enhance palatable food intake, inspired by a protocol previously described[27]. Over the course of two days mice were co-housed with either a Swiss-CD1 aggressor mouse (Stress group) or with a novel C56BL/6 J conspecific (Control). In each case, a transparent semi-permeable barrier was in place between the mice. On each of the two days, once in the morning and once in the evening, the barrier was lifted in the stress group until a total cumulative fighting time of 20 s was reached, after which the separating barrier was placed back. In total, the stressed group thus experienced four episodes of 20 s of fighting time over two days. We confirmed that this procedure was stressful, by assessing anxiety levels on the next day (i.e. post stress day 1; PSD1). On PSD1 stressed mice indeed had fewer open arm entries and head dips in the Elevated Plus Maze (Supplementary Fig. 1a–c) and spent less time in the light compartment of the Light-Dark box (Supplementary Fig. 1d, e).

We next determined whether this social stress paradigm would increase the intake of palatable food (Fig. 1a). In one experiment, on post stress days 1 and 2 (PSD1, PSD2) mice were given ad libitum access to a High Fat High Sugar (HFHS) choice diet, containing separate sources of fat, chow, 10% sucrose water and regular water. Stressed mice exhibited increased intake of both fat and sugar water on PSD1 and PSD2 compared to control (Fig. 1b, c). In contrast, overall chow intake did not differ after stress on PSD1 and decreased on PSD2 (Fig. 1d). Overall caloric intake on PSD1 was increased in the stress group (Fig. 1e), and on PSD2 stressed mice obtained their daily calories to a larger extent from palatable food sources (i.e. fat and sugar) compared to control mice (Fig. 1f). Bodyweight did not change during this brief protocol, not confounding food intake (Supplementary Fig. 1f).

In humans stress can be a potent trigger for binge eating, which is the voracious intake of a large amount of (palatable) food in a short period of time (typically two hours)[28,29]. To determine whether social stress increased binge eating, mice were exposed to two days of social stress or control conditions (Fig. 1a). Then, on PSD1 and PSD2, they were given limited access to fat for 2 h in another cage and otherwise had ad libitum access to chow in their home cage (22 h/day). Social stress increased fat intake overall (Fig. 1g), but did not alter chow intake (Fig. 1h). Also in this assay total caloric intake was increased (Fig. 1i). Overall, these findings indicate that social stress generates enhanced intake of particularly palatable food on subsequent days.

### LHA glutamatergic neurons projecting to the VTA are active during social stress and fat intake
Next, we tested our hypothesis that the LHA$_{glu}$-VTA pathway is sensitive to social stress. LHA$_{glu}$ projections to the VTA have been previously shown to be active during foot shocks[26]. However, it is unclear whether a naturalistic social stressor with established effects on food intake also engages this pathway directly. To examine this we measured the activity of LHA$_{glu}$ neurons projecting to the VTA during fight episodes of the social stress experience. To this end vesicular

glutamate transporter 2-Cre (Vglut2-Cre) mice were stereotactically injected in the VTA with a Herpes Simplex Virus (HSV) driving retrograde Cre-dependent recombination of a genetically encoded calcium indicator (HSV-LS1L-GCaMP6s) in glutamatergic inputs to the VTA. An optic fiber was implanted above the LHA, allowing detection of emitted calcium-driven fluorescence specifically from LHA$_{glu}$ neurons projecting to the VTA (LHA$_{glu}$-VTA) during a fight (Fig. 2a, b). A control group was similarly implanted with a fiber, but injected with an HSV-LS1L-GFP construct in the VTA instead, to control for fight-related non-calcium dependent dynamics (Fig. 2a, b).

We exposed these mice to an aggressive Swiss-CD1 mouse and time-locked recorded fiber photometric signals to the onset of fights. We observed markedly increased fluorescence emitted by LHA$_{glu}$-VTA neurons during fights in the GCaMP but not the GFP group (Fig. 2c, d). To control for aspects of manual scoring of fight onsets, we next used supervised machine learning-based predictive classifiers of fight behavior. We aligned the fiber photometric signals to these algorithm-based detected fights, which confirmed that LHA$_{glu}$-VTA neurons are active during fights (Supplementary Fig. 2a–c). These increased calcium signals during fights could represent the stressful component of this complex naturalistic stimulus, but could also reflect movement or social engagement. To determine whether the locomotor component in a fight contributed to the observed calcium signals, we also recorded from mice alone in a novel environment. We video-detected spontaneously occurring high speed activity bouts and time-locked the onset of these to recorded photometric signals. We did not observe calcium-driven dynamics in the signal linked to the onset of high velocity bouts (Fig. 2e). Next, we determined whether social interaction contributed to the observed calcium signals. We exposed the animals to a juvenile mouse and time-locked recorded signals to the onset of non-hostile social interactions. We did not observe significant calcium signal dynamics linked to the social juvenile interaction (Fig. 2f; Supplementary Fig. 2d). Afterwards we validated that a stressor of an entirely different modality than social stress could also excite the LHA$_{glu}$-VTA pathway. We subjected the mice to electric foot shocks, which indeed increased the activity in LHA$_{glu}$-VTA neurons (Supplementary Fig. 2e). These data suggest that LHA$_{glu}$ neurons projecting to the VTA respond to stressful stimuli, including social fight. Though the optic fibers in this experiment targeted the LHA (Supplementary Fig. 2f), it is still possible that the calcium signals were influenced by non-LHA glutamatergic VTA-projecting cells whose axons or dendrites extend through LHA. To further corroborate that social stress indeed engaged LHA$_{glu}$ cells, we stereotactically injected Vglut2-Cre mice directly in the LHA with an anterograde Cre-dependent vector to drive GCaMP expression (AAV-DIO-GCaMP6s) and placed an optic fiber above it. We exposed these mice to social stress bouts and confirmed that these resulted in strong calcium signals in the LHA$_{glu}$ cellular population (Supplementary Fig. 2g).

We next assessed whether the LHA$_{glu}$ neurons that project to the VTA neurons are activated by fat intake and whether that is altered by our two day social stress paradigm. To establish this we utilized the same approach as before (Fig. 2a). In these animals we measured the GCaMP6s responses of LHA$_{glu}$-VTA neurons during nose pokes into a port with fat, before and after 2 days of social stress (Fig. 2g). We observed that LHA$_{glu}$ cell bodies projecting to the VTA showed calcium transients around interactions with the fat port. Two days of social stress did not alter the magnitude of these responses (Fig. 2h). Together these findings demonstrate that LHA$_{glu}$ neurons projecting to the VTA are activated during aversive social stress experience, as well as during fat intake.

### Stress potentiates LHA$_{glu}$−VTA$_{DA}$ synapses via postsynaptic AMPA receptor modifications
The previous results suggest that LHA$_{glu}$ somata activity during fat intake is not altered by prior social stress. We next tested whether

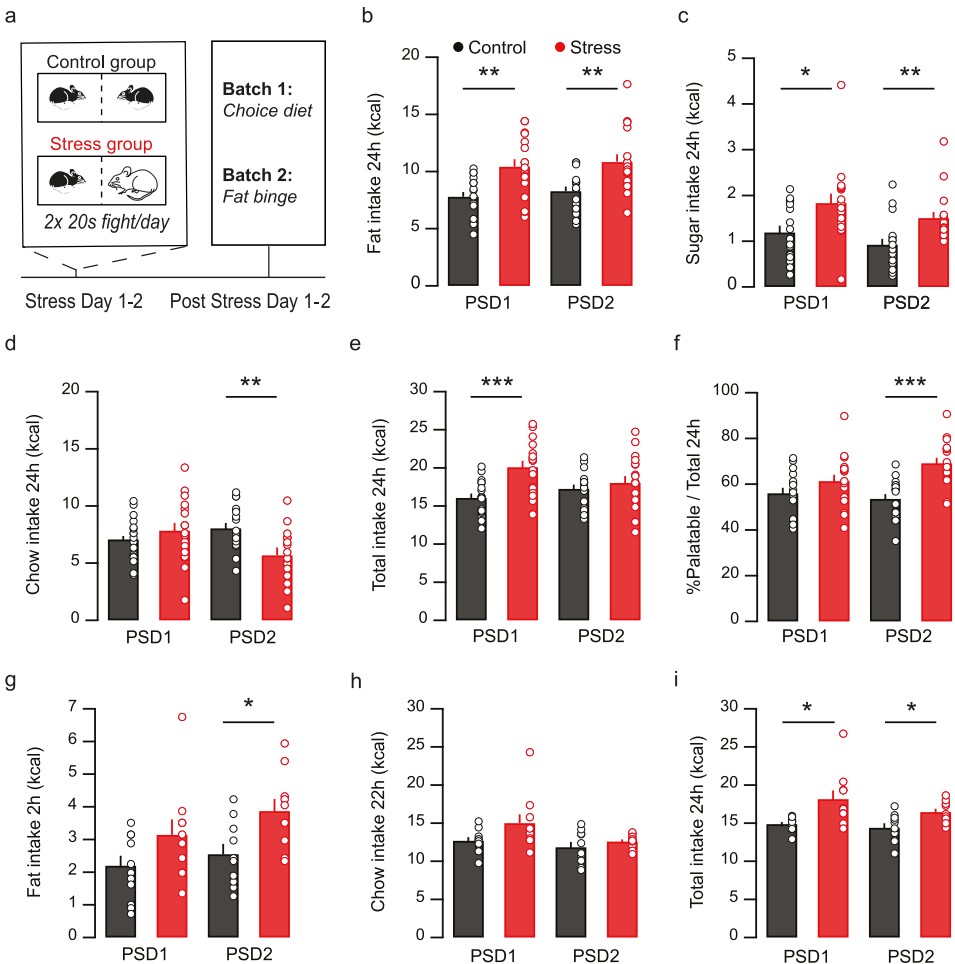

**Fig. 1 | Social stress increases palatable food intake. a** Timeline and schematic for social stress protocol followed by ad libitum High Fat High Sugar (HFHS) diet or limited fat access paradigm (Binge: 2 h fat, 22 h chow). **b** Ad libitum HFHS diet; Fat intake in kcals per day increased post stress day (PSD) 1 and 2 ($n$ group = 17, RM Two-way ANOVA. Main effect stress $F_{(1,32)}$ = 18.44, $p$ = 0.0002; PSD1 $F_{(1,32)}$ = 11.19, $p$ = 0.002; PSD2 $F_{(1,32)}$ = 10.40, $p$ = 0.003). **c** Ad libitum HFHS diet; Sugar water intake in kcals per day increased post stress ($n$ group = 17, RM Two-way ANOVA. Main effect stress $F_{(1,32)}$ = 8.95, $p$ = 0.005; PSD1 $F_{(1,32)}$ = 6.39, $p$ = 0.017; PSD2 $F_{(1,32)}$ = 9.54, $p$ = 0.004). **d** Ad libitum HFHS diet; overall chow intake in kcals did not differ on PSD1 and decreased on PSD2 ($n$ group = 17, RM Two-way ANOVA. Interaction stress x day $F_{(1,32)}$ = 7.41, $p$ = 0.01. PSD1 $F_{(1,32)}$ = 0.84, $p$ = 0.37. PSD2 $F_{(1,32)}$ = 10.34, $p$ = 0.003). **e** Ad libitum HFHS diet; Total caloric intake in kcals increased on PSD1 post stress ($n$ group = 17, Two-way RM ANOVA. Main effect stress $F_{(1,32)}$ = 13.76, $p$ = 0.0008; PSD1 $F_{(1,32)}$ = 15.64, $p$ = 0.0004; PSD2 $F_{(1,32)}$ = 0.52,

$p$ = 0.47). **f** Ad libitum HFHS diet; palatable fat and sugar intake as a percentage from total caloric intake increased on PSD2 ($n$ group = 17, RM Two-way ANOVA. Interaction stress x day $F_{(1,32)}$ = 5.40, $p$ = 0.03; PSD1 $F_{(1,32)}$ = 2.08, $p$ = 0.16; PSD2 $F_{(1,32)}$ = 23.26, $p$ = 0.00003). **g** Limited access paradigm; Caloric intake from fat during 2 h on PSD1 and 2 increased. ($n$ group = 10, RM Two-way ANOVA. Main effect stress $F_{(1,18)}$ = 7.071, $p$ = 0.016; PSD1 $F_{(1,18)}$ = 2.80, $p$ = 0.11; PSD2 $F_{(1,18)}$ = 7.33, $p$ = 0.014). **h** Limited access paradigm; Caloric intake from chow after the 2 h fat exposure on PSD1 and 2 did not differ post stress ($n$ group = 10, RM Two-way ANOVA. Main effect stress $F_{(1,18)}$ = 3.15, $p$ = 0.09). **i** Limited access paradigm; Total caloric intake on PSD1 and 2 increased after stress ($n$ group = 10, RM Two-way ANOVA. Main effect stress $F_{(1,18)}$ = 12.18, $p$ = 0.0026; PSD1 $F_{(1,18)}$ = 7.75, $p$ = 0.012; PSD2 $F_{(1,18)}$ = 7.84, p = 0.011). All data are shown as mean + SEM and statistical tests were performed two-sided. *$p < 0.05$; **$p < 0.01$; ***$p < 0.001$. Source data are provided as a Source Data file.

social stress would instead affect synaptic strength of LHA$_{glu}$ neurons onto VTA cell types. We hypothesized that social stress would cause potentiation of LHA$_{glu}$-VTA$_{DA}$ synapses. To test this, Pitx3-GFP mice (with green fluorescent DA neurons) were stereotactically injected in the LHA with an Adeno-Associated Viral vector driving the expression of the light-activatable cation channel Channelrhodopsin2 (ChR2) (AAV5-hSyn-ChR2(H134R)-mCherry). After a recovery and viral incubation period, we exposed the mice to the two-day social stress protocol (or its control) and sacrificed the mice on PSD1 (i.e. at a moment were they would start overconsuming palatable foods if available; Fig. 1b–i). We prepared brain slices and performed ex vivo patch clamp recordings of fluorescently labelled VTA$_{DA}$ neurons (in the absence of tetrodotoxin for polysynaptic block) for optogenetic-assisted circuit interrogation (Fig. 3a, b). To determine LHA glutamatergic synaptic strength onto VTA dopamine neurons, we assessed the ratio of AMPAR-mediated currents (recorded at −65 mV) to

NMDAR-mediated ones (at + 40 mV, 100 ms after stimulation) (Supplementary Fig. 3a, b). Two days of social stress increased the AMPAR-NMDAR ratio at LHA$_{glu}$-VTA$_{DA}$ synapses, suggesting increased excitatory synaptic strength (Fig. 3c). Enhanced AMPAR-NMDAR ratio was also observed when both synaptic components were measured at depolarized potentials (+40 mV) and pharmacologically isolated (Supplementary Fig. 3c). This hints towards an overall increased number of postsynaptic AMPARs post stress. Another potential contributing factor to an increased AMPAR-NMDAR ratio is a switch in postsynaptic membrane AMPAR subunit composition, from receptors with GluA2 subunits to GluA2-lacking ones[30]. In accordance with this scenario, we observed that social stress increased AMPAR inward rectification, with proportionally larger AMPAR currents detected at hyperpolarized than at depolarized potentials (Fig. 3d). As opposed to the stress-driven postsynaptic changes at LHA$_{glu}$-VTA$_{DA}$ synapses, we did not observe

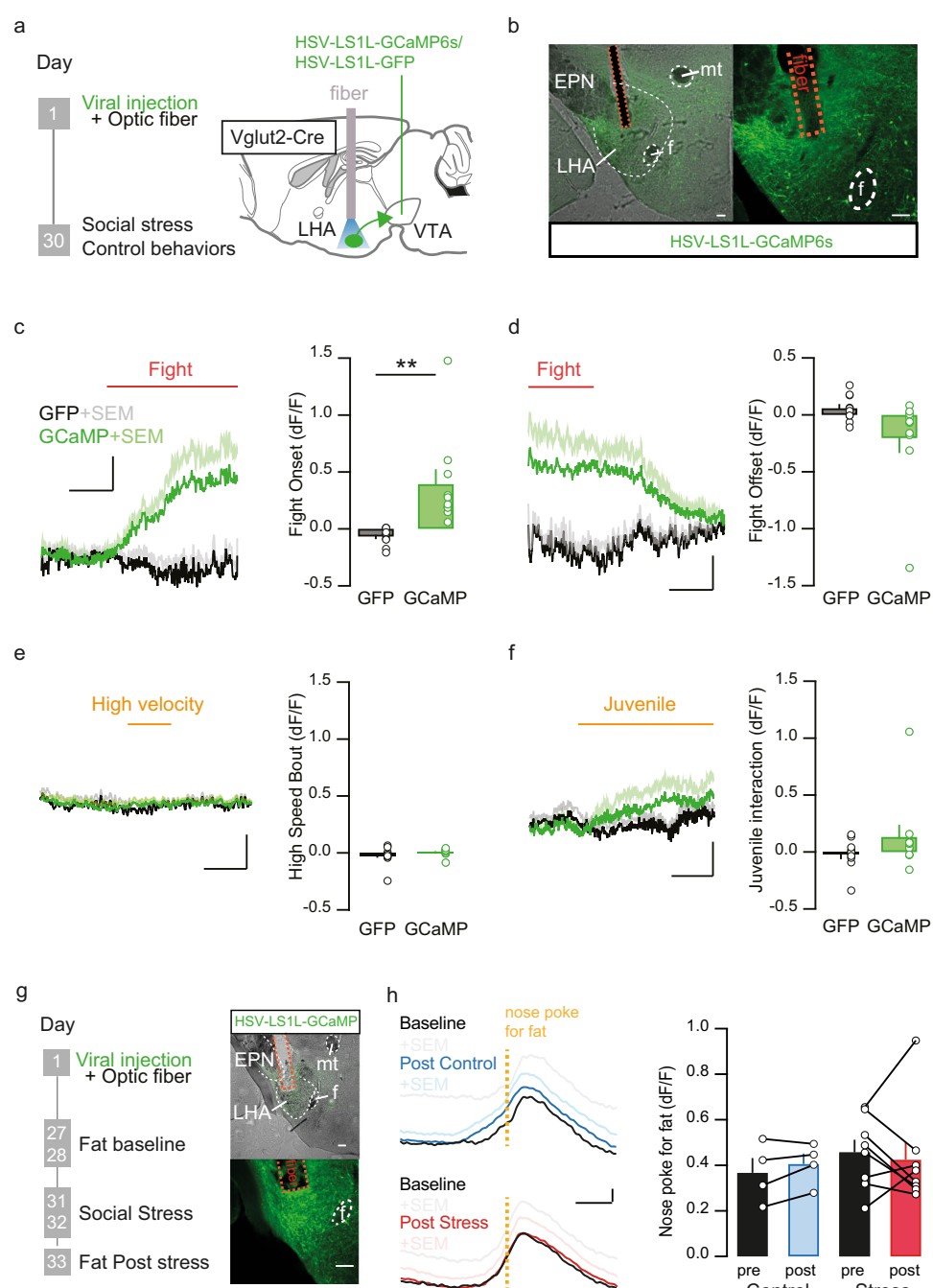

**Fig. 2 | LHA glutamatergic neurons projecting to VTA are stress sensitive.**
**a** Timeline and schematic for Fiber photometry experiments during stress.
**b** Representative image of viral expression of GCaMP6s in LHA$_{glu}$ neurons projecting to the VTA (in green), with an optic fiber above the LHA (scale bars: 100 μm). Anatomical reference points: f fornix, EPN entopeduncular nucleus, mt mammillothalamic tract. **c** Average recorded emitted fluorescent signal of GCaMP6s with SEM (dF/F) (green; $n = 10$) and control GFP (black; $n = 10$) of LHA$_{glu}$ neurons projecting to VTA, time-locked to fights (red line). Scale bars: 2 s, 0.25 dF/F. Quantified peak signals compared to baseline during fight onset of GCaMP and GFP control groups with SEM (mice per group $n = 10$, One-Way ANOVA $F_{(1,18)} = 12.03$, $p = 0.003$). **d** As (**c**), but for fight offset (end of the red line) (mice per group $n = 10$, One-Way ANOVA, $F_{(1,18)} = 3.65$, $p = 0.072$). **e** Average recorded emitted fluorescent signal plus SEM of GCaMP6s signal and GFP (dF/F) of LHA$_{glu}$-VTA neurons time-locked to the onset of high velocity bouts. Scale bars: 1 s, 0.25 dF/F. Quantified peak signals compared to baseline during high velocity bouts for GCaMP and GFP control groups. Error bars indicate SEMs (mice per group $n = 9$, One-Way ANOVA,

$F_{(1,16)} = 1.00$, $p = 0.33$). **f** Average recorded emitted fluorescent signal plus SEM of GCaMP6s signal and GFP (dF/F) of LHA$_{glu}$ neurons projecting to VTA time-locked to juvenile interactions. Scale bars: 1 s, 0.25 dF/F. Quantified peak signals compared to baseline during juvenile interaction for GCaMP and GFP control groups. Error bars indicate SEMs (mice per group $n = 10$, One-Way ANOVA, $F_{(1,18)} = 1.75$, $p = 0.20$). **g** Timeline and schematic for fiber photometry experiments during nose pokes for fat. On the right same as (**b**). **h** Left: Average recorded emitted fluorescent signal of GCaMP6s plus SEM (dF/F) (pre and post control/stress, respectively black and blue; $n = 4$) and stress (pre and post stress, respectively black and red; $n = 8$) of LHA$_{glu}$ neurons projecting to VTA, time-locked to the onset of nose pokes for fat (dashed orange line). Scale bars: 1 s, 0.1 dF/F. Right: Quantified peak signals compared to pre nose poke during fat nose poke of control and stress animals pre and post stress +SEM (RM Two-way ANOVA. Main effect stress $F_{(1,10)} = 0.338$, $p = 0.54$; main effect time $F_{(1,10)} = 0.004$, $p = 0.95$; main interaction effect time*stress, $F_{(1,10)} = 0.535$, $p = 0.48$) . All statistical tests were performed two-sided. **$p < 0.01$. Source data are provided as a Source Data file.

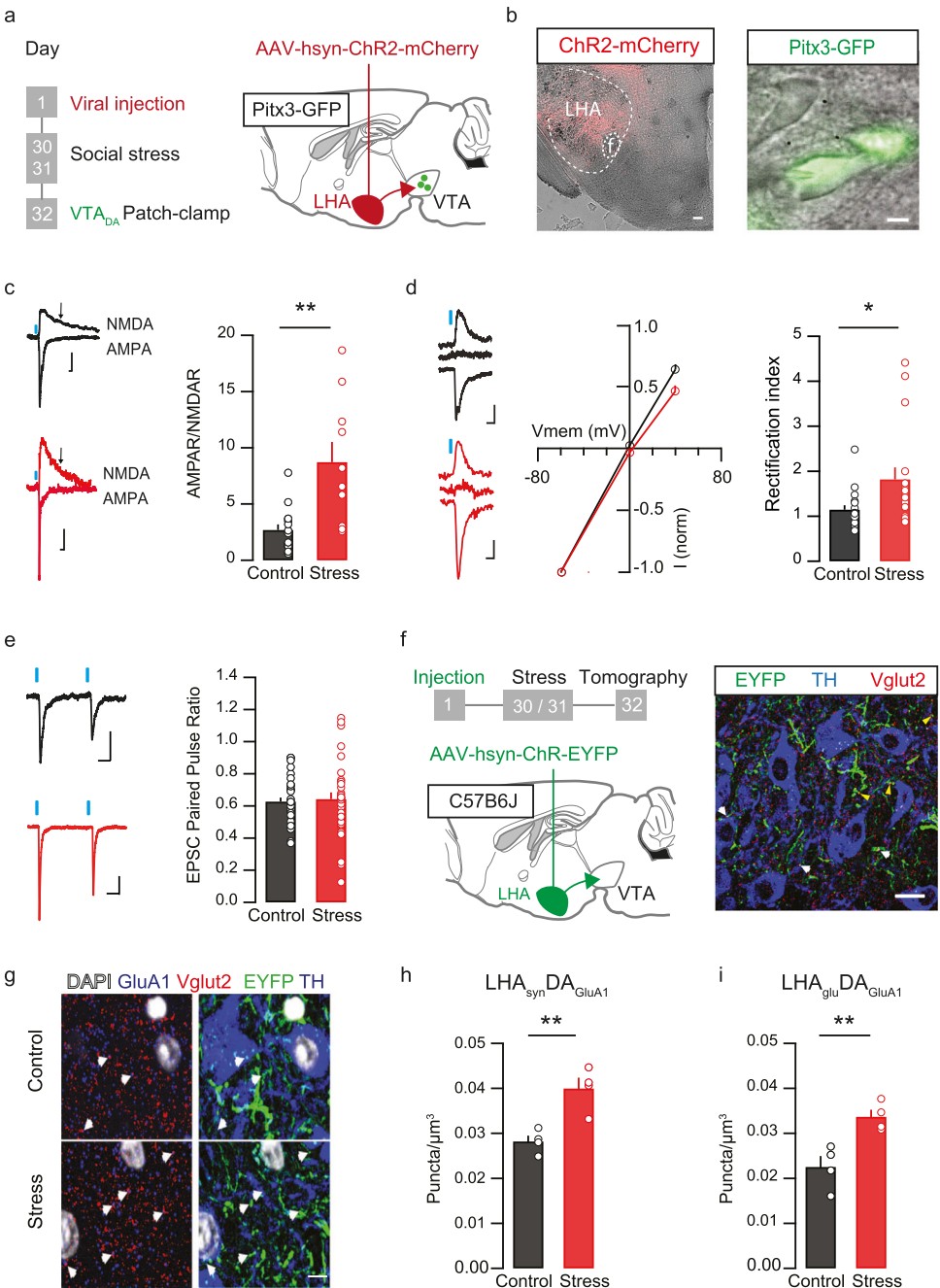

**Fig. 3 | Social stress potentiates LHA$_{glu}$-VTA$_{DA}$ synapses via AMPA receptor modifications. a** Timeline and schematic for electrophysiological patch clamp experiments. **b** Representative image of an LHA slice with ChR2-mCherry in red. Scale bar: 100 μm. Anatomical reference point: f fornix. Right: Image of green fluorescent VTA$_{DA}$ neurons (Pitx3-GFP) with patch clamp electrode. Scale bar: 10 μm. **c** AMPAR-NMDAR ratios at LHA$_{glu}$-VTA$_{DA}$ synapses increase post stress. Left: representative I-AMPAR and I-NMDAR traces, Blue rectangle: start of LED stimulation. Scale bar: 20 ms, 20 pA. I-AMPAR measured at −65 mV, I-NMDAR at +40 mV at 100 ms from stimulation onset (black arrows). Right quantification of AMPAR/NMDARs +SEMs (*n* cells control = 15, *n* cells stress = 10, One-Way ANOVA, $F_{(1,23)} = 14.72$, $p = 0.001$). **d** AMPAR Rectification index increases at LHA$_{glu}$-VTA$_{DA}$ synapses after stress. Left: example traces, scale bars: 20 ms, 10 pA. Middle: AMPAR current at −60 mV, 0 mV and +40 mV normalized to AMPAR current at −60 mV (*n* cells control = 18, *n* cells stress = 17). Right: Averages + SEMs (*n* cells control = 18, *n* cells stress = 17, One-Way ANOVA, $F_{(1,33)} = 5.81$, $p = 0.02$). **e** Paired pulse ratio of LHA$_{glu}$-VTA$_{DA}$ neurons does not differ due to stress. Left: example traces. Scale bars: 25 pA, 25 ms. Right: Averages + SEMs (*n* cells = 31 per group, One-Way ANOVA, $F_{(1,60)} = 0.02$, $p = 0.89$). **f** Left: Schematic for Array Tomography experiment.

Right: Representative image of VTA slice stained for LHA fibers (YFP: green), DA neurons (TH: blue) and glutamatergic fibers (Vglut2: red). White arrows: puncta overlap between YFP, Vglut2 and TH. Yellow arrows: puncta with overlap between YFP and Vglut2. Scale bar: 10 μm. **g** Representative images of reconstructed VTA slices with array tomography. On the left side the white arrows indicate overlap between Vglut2+ glutamatergic terminals (red) with GluA1 subunits (blue). On the right side the same slice is shown, where the white arrows indicate overlap between LHA terminals (YFP) and dopamine neurons (TH: blue). In both cases DAPI-stained nuclei are shown (white). The top row shows an example for a control mouse, and the bottom row for a stressed mouse. Scale bar: 5 μm. **h** Quantification of overlap of LHA terminals, synapsin, GluA1 AMPAR subunits and dopamine neurons, for Control and Stress conditions (averages + SEMs, mice per group *n* = 4, One-Way ANOVA, $F_{(1,6)} = 18.26$, $p = 0.005$). **i** As h but for the overlap of LHA terminals, Vglut2, GluA1-AMPAR subunits and dopamine neurons for Control and Stress conditions (averages + SEMs, mice per group *n* = 4, One-Way ANOVA, $F_{(1,6)} = 14.9$, $p = 0.008$). All statistical tests were performed two-sided *$p < 0.05$, **$p < 0.01$. Source data are provided as a Source Data file.

changes in presynaptic neurotransmitter release probability, as measured by the paired-pulse ratio (PPR) (Fig. 3e).

To further validate our functional synaptic data, we sought to provide structural insights into effects of stress on LHA$_{glu}$-VTA$_{DA}$ synaptic connectivity. To this end we performed quantitative high-resolution array tomography of LHA$_{glu}$ synapses onto VTA$_{DA}$ neurons. We injected C56BL/6 J mice with an anterograde tracer (AAV5-hSyn-ChR2-EYFP) in the LHA. These mice were subjected to two days of social stress and we perfused and fixated the brains on PSD1. Thereafter, the same serial-ultrathin (100 nm) VTA sections were immuno-labeled against multiple synaptic markers. This allowed us to quantify LHA synapses specifically onto VTA$_{DA}$ neurons (Fig. 3f, g). In stressed mice, we found more associations between LHA synapses (EYFP + axon boutons co-labeled for synapsin) and VTA$_{DA}$ neuronal postsynaptic GluA1-AMPAR subunits (GluA1 + puncta associated with TH + cells) (Fig. 3h). We confirmed that this increase in synaptic associations was also observed when specifically evaluating glutamatergic presynaptic LHA terminals (EYFP + axon boutons co-labeled for Vglut2) with GluA1+ puncta onto VTA$_{DA}$ neurons (Fig. 3i). These increased associations occurred despite there being no general stress-driven changes in overall LHA-to-VTA anatomical innervation (Supplementary Fig. 3d), or changes in total VTA GluA1-AMPAR levels (Supplementary Fig. 3e). Overall, these data suggest that LHA$_{glu}$-VTA$_{DA}$ synapses are potentiated by social stress due to an increase in the number of postsynaptic GluA-1 containing AMPARs.

## Social stress decreases inhibition/excitation balance at LHA synapses on VTA$_{DA}$ but not VTA$_{GABA}$ neurons

We next determined the consequences of stress-strengthened LHA$_{glu}$-VTA$_{DA}$ synapses for VTA circuit function. The LHA sends both glutamatergic and GABAergic projections to the VTA, impinging not only on VTA$_{DA}$ but also on VTA$_{GABA}$ (inter)neurons[20–22,31]. Therefore we sought to more integrally understand the consequence of LHA$_{glu}$-VTA$_{DA}$ synaptic plasticity for the VTA$_{DA}$ system. We first addressed whether the observed changes at LHA$_{glu}$-VTA$_{DA}$ synapses would shift the overall balance between excitatory and inhibitory LHA synaptic input, and whether this would specifically occur in VTA$_{DA}$ over VTA$_{GABA}$ neurons. To that end, to record from LHA-VTA$_{DA}$ pathways we again virally expressed ChR2 in the LHA of Pitx3-GFP mice (Fig. 4a, b). Instead, to record from LHA-VTA$_{GABA}$ pathways, we injected vesicular GABA transporter (VGAT)-Cre mice with the same AAV vector to target ChR2 expression in the LHA, along with a Cre-dependent fluorophore in the VTA (AAV5-EF1a-DIO-EYFP; Fig. 4a). In either case we recorded opto-stimulated AMPA receptor (AMPAR)- and GABA$_A$ receptor (GABA$_A$R)-dependent synaptic currents at LHA-VTA synapses at −65 mV and 0 mV, respectively (Supplementary Figs. 3a, 4a).

For VTA$_{DA}$ neurons we observed that 81% of patched cells responded with at least excitatory postsynaptic currents to LHA optical stimulation (Supplementary Fig. 4b, c). Of those VTA$_{DA}$ neurons that responded to LHA stimulation, 41% received only glutamatergic input, 47% received both glutamatergic and GABAergic input, and 12% received only GABAergic input (Supplementary Fig. 4d). The GABA$_A$R-AMPAR ratio was calculated to assess the inhibition-excitation balance of overall LHA input to VTA$_{DA}$ neurons. VTA$_{DA}$ neurons showed reduced GABA$_A$R-AMPAR ratios after two days of social stress (Fig. 4b). For VTA$_{GABA}$ neurons the responsivity to LHA opto-stimulation with at least excitatory currents was 84%, comparable to the proportion of VTA$_{DA}$ neurons connected to the LHA (Supplementary Fig. 4c). Of the responsive VTA$_{GABA}$ neurons 27% received only AMPAR, 53% both AMPAR and GABA$_A$R and 20% only GABA$_A$R LHA input (Supplementary Fig. 4d). In contrast to the VTA$_{DA}$ neurons, the GABA$_A$R-AMPAR ratio of LHA inputs to VTA$_{GABA}$ neurons was not altered by the social stress paradigm (Fig. 4c). The probability of presynaptic neurotransmitter release from LHA$_{glu}$ or LHA$_{GABA}$ terminals onto VTA cellular populations was not affected by

stress, as measured by paired pulse ratios (Supplementary Fig. 4e–f; Fig. 3e).

Overall, in combination with our previous findings (Fig. 3c, d, h, i), this suggests that the overall net excitatory drive from the LHA is increased onto VTA$_{DA}$ neurons after stress. In further accordance with this, we observed an increased amplitude of (non-projection specific) glutamatergic spontaneous excitatory postsynaptic currents (sEPSCs) onto VTA$_{DA}$ neurons after stress (Supplementary Fig. 4g). Instead, the frequency of sEPSCs and the frequency and amplitude of GABAergic spontaneous inhibitory postsynaptic currents (sIPSCs) onto VTA$_{DA}$ neurons did not differ after stress (Supplementary Fig. 4g, h). Together these findings suggest a stress-driven increase in the balance of excitatory versus inhibitory LHA inputs onto VTA$_{DA}$ but not onto VTA$_{GABA}$ neurons at a time point coinciding with enhanced intake of palatable food.

## Social stress potentiates LHA$_{glu}$-VTA$_{DA}$-mPFC but not LHA$_{glu}$-VTA$_{DA}$-medial NAc synapses

VTA$_{DA}$ neurons have different output targets, which include the NAc medial shell and the medial prefrontal cortex (mPFC), both known to be implicated in reward processing and to be acutely stress sensitive[5,11,26,32]. Previous work has shown that painful stimuli potentiate glutamatergic synapses of unknown origin onto VTA$_{DA}$ neurons projecting to the mPFC but not to the medial NAc shell (NAc mshell)[17]. We therefore sought to determine whether social stress would preferentially potentiate LHA glutamatergic synapses onto mPFC-projecting versus NAc mshell-projecting VTA dopamine neurons. To address this, we injected Pitx3-Cre mice with a Cre-dependent retrograde tracer in the NAc mShell (GFP) and the mPFC (mCherry) in the same mouse, in addition to driving ChR2 expression in the LHA (Fig. 4d, Supplementary Fig. 5a). We then subjected these mice to the two day social stress protocol, and made brain slices the day after. We patched from fluorescently tagged VTA dopamine neurons and measured LHA glutamatergic rectification properties as a measure for glutamatergic strength (Fig. 3d). We observed that social stress increased the rectification index at LHA$_{glu}$ synapses onto VTA$_{DA}$-mPFC neurons (Fig. 4e), but not at LHA$_{glu}$ synapses onto VTA$_{DA}$-NAc mshell neurons (Supplementary Fig. 5b). This finding is in accordance with a stress-driven potentiation of LHA glutamatergic synapses at mPFC-projecting VTA$_{DA}$ neurons.

## Social stress enhances phasic LHA-driven VTA$_{DA}$ output in the mPFC

We next addressed if these synaptic alterations after stress would have consequences for how the LHA controls VTA$_{DA}$ output. First, we investigated whether the basal intrinsic properties of VTA$_{DA}$ neurons themselves (i.e. in the absence of coordinated phasic LHA input) were altered by our stress protocol. We performed cell-attached and whole-cell current clamp recordings in VTA$_{DA}$ neurons in brain slices from Pitx3-GFP mice. We observed that in the absence of coordinated phasic LHA input, VTA$_{DA}$ neuronal spontaneous firing frequency, excitability, voltage sag and membrane resistance were not affected by the social stress protocol (Supplementary Fig. 4i–l).

We hypothesized that since social stress potentiates glutamatergic synapses on VTA$_{DA}$ rather than VTA$_{GABA}$ neurons (Fig. 4b, c) without affecting intrinsic properties of VTA$_{DA}$ neurons (Supplementary Fig. 4i–l), phasic LHA$_{glu}$-VTA stimulation after stress would drive more dopamine release in key VTA output regions. We first evaluated the dopamine output to the NAc mshell. We injected Vglut2-Cre mice with a Cre-dependent high-conducting variant of ChR2 (Chloromonas oogama Channelrhopdopsin (CoChR); AAV5-Flex-CoChR−GFP)[33] in the LHA and implanted an optic fiber above the VTA to stimulate LHA$_{glu}$-VTA synapses. In addition, we virally targeted dopamine biosensor dLight1.1[34] in the NAc mshell and placed an optic fiber above it to detect dopamine transients there

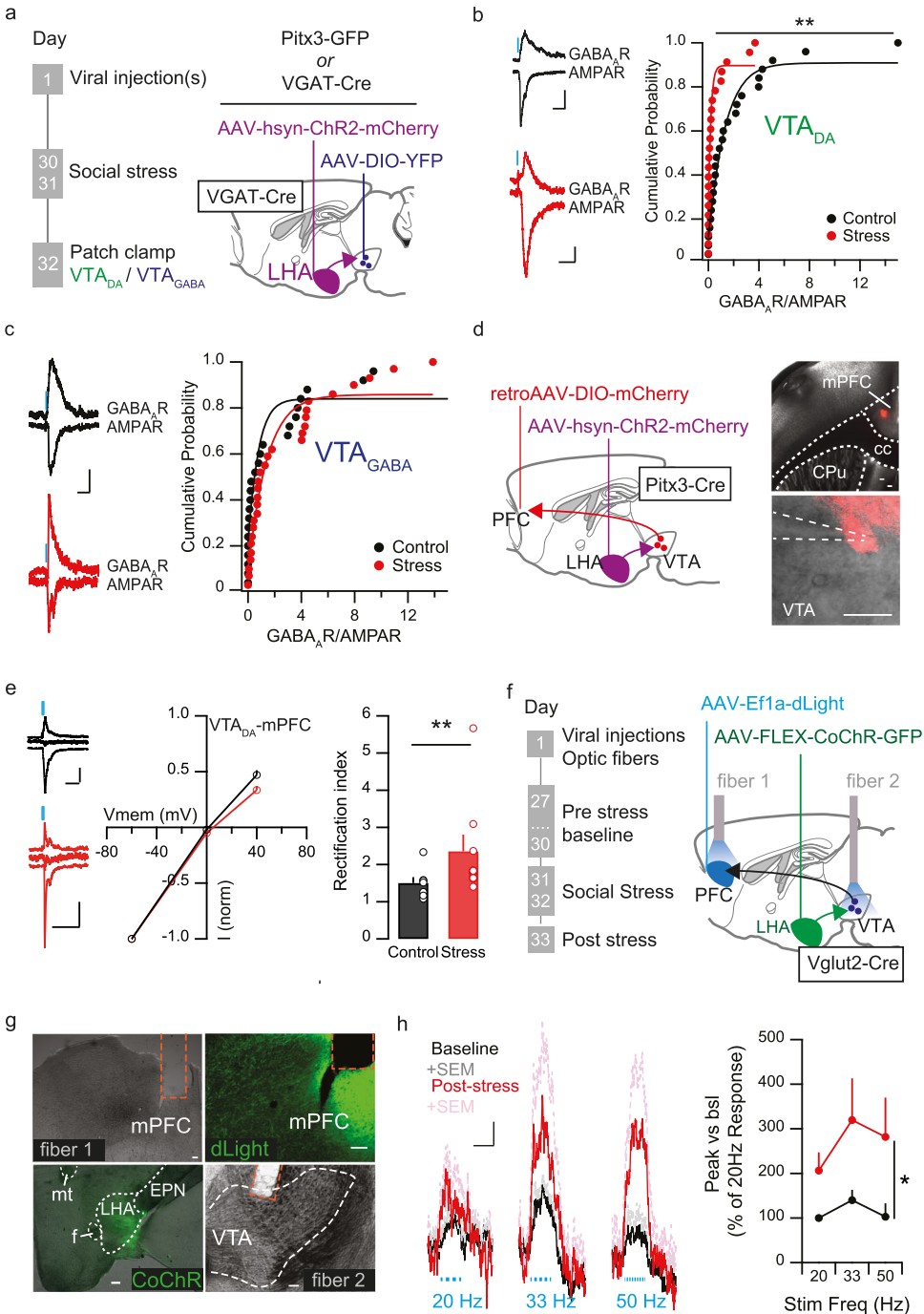

**Fig. 4 | Social stress shifts the balance of excitation and inhibition more towards excitation at LHA control over the VTA dopamine system. a** Timeline and schematic for stress protocol for electrophysiological patch clamp experiments of VTA_DA and VTA_GABA neurons. **b** GABA_AR-AMPAR ratios at LHA-VTA_DA synapses decreased post stress. Left: representative AMPAR and GABA_AR traces. Blue rectangle: start of LED stimulation. Scale bar: 25 pA, 20 ms. Right: Cumulative probability plot quantification (*n* cells control = 23, *n* cells stress = 25, KS-test, *D* = 0.499, *p* = 0.003). **c** GABA_AR-AMPAR ratio at LHA-VTA_GABA synapses. Left: representative AMPAR and GABA_AR traces. Scale bar: 40 pA, 25 ms. Right: Cumulative probability plot quantification (*n* cells control = 25, *n* cells stress = 29, KS-test, *D* = 0.34, *p* = 0.074). **d** Left: Schematic of patch clamp experiments of VTA_DA neurons projecting to mPFC. Right: representative image of viral expression of retroAAV-DIO-mCherry (+ diluted red beads for injection site verification) in horizontal sections. mPFC (top) Scale bar: 100 μm, VTA (bottom) scale bar: 20 μm. Anatomical reference points cc corpus callosum, CPu caudate putamen. **e** AMPAR Rectification index increases at LHA_glu-VTA_DA-mPFC synapses after stress.

Left: example traces, scale bars: 50 ms, 25 pA. Middle: AMPAR current at −60 mV, 0 mV and +40 mV normalized to AMPAR current at −60 mV (*n* cells control = 7, *n* cells stress = 9). Right: Averages + SEMs (*n* cells control = 7, *n* cells stress = 9, KS-test, D_(1,15) = 1.70, *p* = 0.006). **f** Timeline and schematic for in vivo fiber photometry experiment to assess mPFC dopamine release as a function of LHA_glu-VTA stimulation. **g** Representative images of CochR-GFP virus expression in LHA_glu cells (green), optogenetic stimulation fiber above the VTA, dLight expression in the PFC (green) with a photometry fiber. Scale bars: 100 μm. Anatomical reference points: f fornix, EPN entopeduncular nucleus, mt mammillo-thalamic tract. **h** Left: Average +SEM of emitted fluorescence in mPFC for LHA_glu-VTA stimulation of 20, 33 and 50 Hz for pre and post stress (*n* mice = 6). Right: Averages + SEMs. Stress increases dopamine release in mPFC post-stress, peak DA release (RM Two-Way ANOVA. Main effect stress F_(1,5) = 6.92, *p* = 0.047). Scale bars: 10 s, 0.05 dF/F. All statistical tests were performed two-sided. *p* < 0.05, **p* < 0.01. Source data are provided as a Source Data file.

(Supplementary Fig. 5c). Thereafter we established an input-output curve stimulating $LH_{glu}$-VTA synapses with trains of 10 or 20 Hz[26]. Concurrently we recorded NAc dopamine dynamics elicited by $LHA_{glu}$-VTA opto-stimulation and established that the resultant dopamine signals were stable over baseline days (Supplementary Fig. 5d, e). We then exposed the mice to the two-day stress protocol, and on PSD1 opto-stimulated $LHA_{glu}$-VTA terminals as before. We observed that the stress experience did not affect NAc mshell dopamine output during $LHA_{glu}$-VTA stimulation compared to baseline, across stimulation frequencies (Supplementary Fig. 5f).

We next tested the influence of social stress on $LHA_{glu}$-VTA driven dopamine release in the mPFC. We had the same viral strategy as described for the NAc mshell, except for the expression of dopamine biosensor dLight1.1 in the mPFC and an optic fiber placement above it to detect dopamine transients there (Fig. 4f, g). As dopamine levels are less abundant in mPFC than in NAc, we used burst-like stimulation patterns at higher frequencies, inspired by previous studies[35]. We established an input-output curve of opto-stimulation trains comprising bursts of 20, 33 or 50 Hz. Also for the mPFC we could detect dopamine dynamics elicited by opto-stimulation which were stable over baseline days (Supplementary Fig. 5g). We then exposed the mice to the two-day stress protocol, and on PSD1 again opto-stimulated $LHA_{glu}$-VTA terminals. We observed that the stress experience increased mPFC dopamine output during $LHA_{glu}$-VTA stimulation compared to baseline, across stimulation frequencies (Fig. 4h). Together these data show that social stress potentiates $LHA_{glu}$-VTA$_{Da}$ synapses, shifting the overall balance of synaptic communication between LHA and VTA more towards LHA-driven excitation of VTA$_{DA}$ neurons, and enhancing the ability of LHA to drive phasic VTA$_{DA}$ output to regions like mPFC.

These data suggest that social stress can alter $LHA_{glu}$-VTA driven dopamine dynamics in the mPFC. Interestingly, previous work has shown that the dopamine 1 receptor (DRD1) expressing population of mPFC neurons contributes to food intake[32]. We asked whether such dopamine-sensitive mPFC$_{D1R}$ cells could contribute to fat binge behavior in ad libitum fed mice. To mimic the stimulatory effect dopamine can have on mPFC$_{D1R}$ neurons[36], we expressed a stimulatory DREADD (rAAV5-hSyn-DIO-hM3D(Gq)-mCherry) or a control fluorophore in the DRD1 expressing neurons in the mPFC (Supplementary Fig. 5h). When stimulating these neurons with DREADD agonist Compound 21 (C21), fat intake was indeed increased (Supplementary Fig. 5i). While it is still plausible that multiple VTA output pathways contribute to stress eating responses, these findings are in accordance with a link between stress-altered dopamine signaling in mPFC and increased palatable food intake.

## Mimicking effects of social stress on $LHA_{glu}$-VTA pathway leads to increased fat intake

We next investigated whether there is a causal connection between the stress-induced potentiation of $LHA_{glu}$-VTA$_{DA}$ synapses and altered food intake. We predicted that mimicking the effect of stress on these synapses would also increase palatable fat intake. Previous research has shown that glucocorticoid receptor (GR) stimulation can potentiate glutamatergic afferents, of unknown origin, onto VTA$_{DA}$ neurons[15,16]. To determine whether this could be a relevant induction mechanism for potentiation of $LHA_{glu}$-VTA synapses we first established that the main endogenous GR agonist in mice, corticosterone, was indeed elevated by bouts of social stress (Supplementary Fig. 6a). We next assessed whether exposing brain slices to a GR agonist would be sufficient to potentiate $LHA_{glu}$-VTA$_{DA}$ synapses. We again virally expressed ChR2 in the LHA and recorded from VTA$_{DA}$ neurons in naïve Pitx3-GFP mice of which slices had been pre-incubated in either the GR agonist dexamethasone (Dex) or its vehicle for 30 min. Dex incubation indeed potentiated $LHA_{glu}$-VTA$_{DA}$ synapses, as observed by an increase in maximal opto-stimulated AMPAR amplitudes (Supplementary

Fig. 6b; Fig. 5a). In addition Dex incubation also resulted in increased AMPAR rectification at $LHA_{glu}$-VTA$_{DA}$ synapses, similar to a two day social stress protocol (Supplementary Fig. 6c, d). Having confirmed that Dex mimics stress-driven potentiation of $LHA_{glu}$-VTA$_{DA}$ synapses ex vivo, we assessed whether infusing Dex in the VTA mimicked stress-driven fat intake as well. We implanted C56BL/6 J mice with bilateral cannulas above the VTA and locally infused Dex or vehicle. Thirty minutes later the animals were given 2 h access to fat, followed by 22 h access to chow (i.e. the limited fat access model, which allows for clearly temporally restricted feeding; Fig. 1a, g–i). The day after they were again given access to 2 h fat followed by 22 h chow without any additional infusions (Fig. 5b). Fat intake was significantly higher in the Dex group both subsequent days, while there was no difference in chow intake (Fig. 5c, Supplementary Fig. 6e). This suggests that social stress can elevate corticosterone levels which, via GR-signalling potentially in the VTA, can enhance $LHA_{glu}$-VTA$_{DA}$ synapses and increase palatable fat intake.

We sought to further corroborate a link between the effects of social stress on $LHA_{glu}$-VTA circuitry and palatable fat intake. Since we observed that $LHA_{glu}$ neurons projecting to the VTA are activated during social stress (Fig. 2c), we investigated whether direct bursts of high frequency stimulation (HFS) of this pathway in non-stressed mice would mimic effects of stress on fat intake. To this end Vglut2-Cre animals were injected in the LHA with Cre-dependent CoChR (AAV5-Syn-FLEX-CoChR-GFP) or a control vector (AAV5-EF1a-DIO-EYFP) and optic fibers were bilaterally implanted above the VTA. After a recovery period, we in vivo optogenetically stimulated $LHA_{glu}$-VTA synapses on two subsequent days for 10 min/day with 20 Hz bursts (5 s on, 10 s off) (Supplementary Fig. 6f, g). Thereafter we measured the chow and fat intake during 2 h. We observed that HFS indeed increased fat intake, while chow intake was not altered (Supplementary Fig. 6h, i). Overall these findings show that mimicking effects of social stress on the $LHA_{glu}$-VTA pathway recapitulates effects of social stress on palatable fat intake.

## Weakening potentiated $LHA_{glu}$-VTA synapses reduces stress-induced fat intake

We next assessed whether stress-driven strengthening of $LHA_{glu}$ input to the VTA is not only sufficient but also necessary for stress-driven increases in fat intake. To that end we sought to reduce $LHA_{glu}$-VTA$_{DA}$ synapse strength in stressed mice. Synaptic strength can often be longitudinally reduced by low frequency stimulation (LFS)[37]. We therefore first tested in brain slices whether LFS could diminish stress-potentiated $LHA_{glu}$-VTA$_{DA}$ synapses. Pitx3-GFP mice expressing ChR2 in the LHA were subjected to two days of social stress and were sacrificed the next day when we performed patch-clamp recordings of VTA$_{DA}$ neurons. After obtaining a stable baseline of opto-evoked glutamatergic synaptic responses at $LHA_{glu}$-VTA$_{DA}$ synapses, we applied a 1 Hz LFS synaptic stimulation protocol for 10 min and evaluated synaptic response strength afterwards. Both control and stressed animals showed a strong reduction of EPSCs after LFS, lasting at least 30 min (Supplementary Fig. 7a). This confirmed that 1 Hz LFS is a useful tool to reduce synaptic strength at $LHA_{glu}$-VTA$_{DA}$ synapses. Next, we tested whether an LFS protocol applied in vivo in freely moving mice would also diminish synaptic strength. Vglut2-Cre animals were injected in the LHA with Cre-dependent CoChR (AAV5-Syn-FLEX-CoChR-GFP) and optic fibers were bilaterally implanted above the VTA. This enabled us to specifically stimulate $LHA_{glu}$ terminals in the VTA in vivo. These animals were exposed to either the two-day control or stress protocol. The next day the stressed animals were given either 30 min mock stimulation or 30 min 1 Hz LFS, and were thereafter prepared for ex vivo patch clamp experiments of VTA neurons (Fig. 5d, e). We observed that opto-evoked $LHA_{glu}$ inputs onto VTA neurons were significantly reduced in the Stress + 1 Hz stim group compared to the Stress + Mock stim group (Fig. 5f). This confirmed

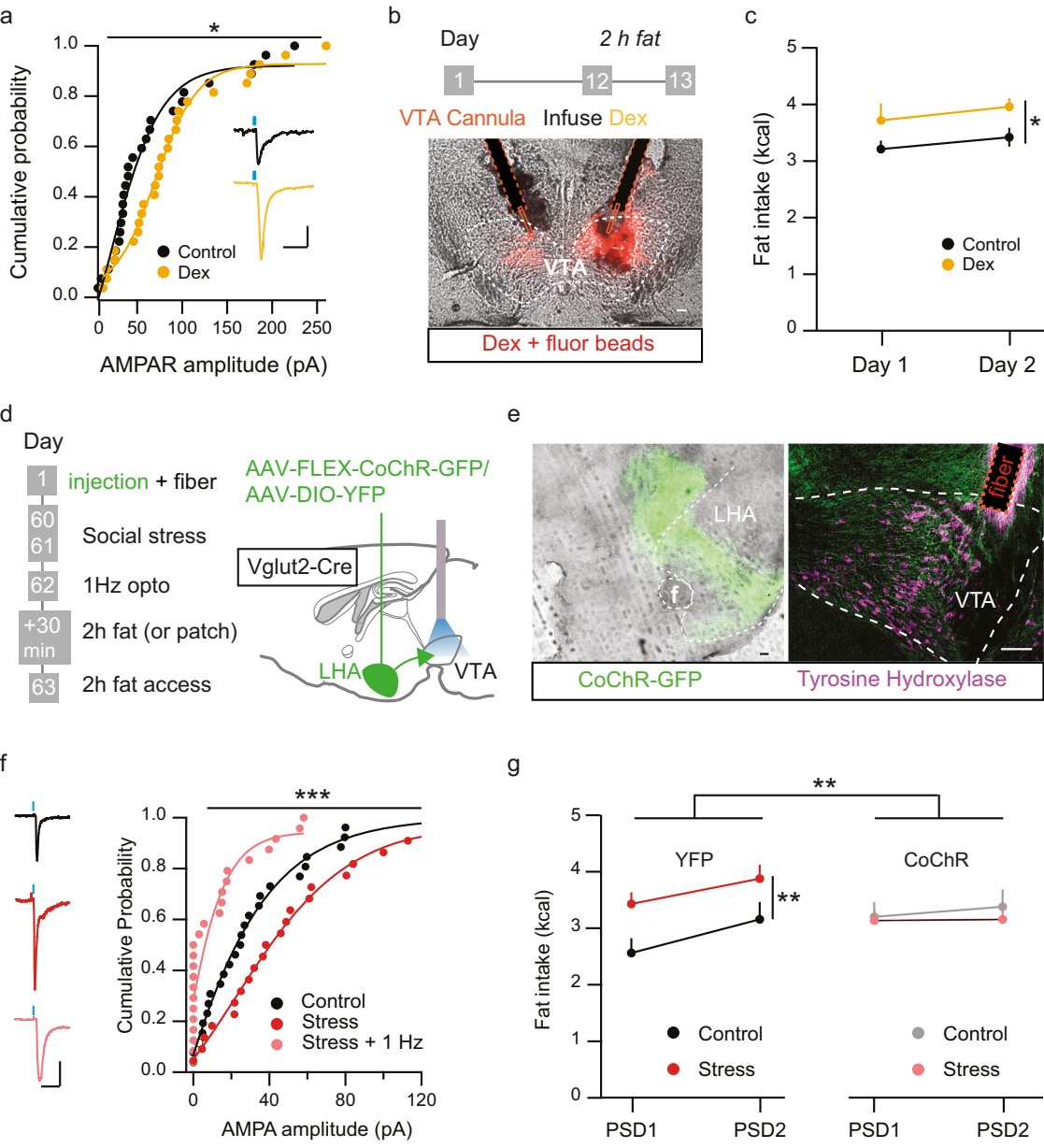

**Fig. 5 | Strengthening or weakening of LHA_glu-VTA synaptic transmission respectively mimics and prevents stress-driven fat intake in a limited access model. a** Opto-evoked AMPAR amplitudes at LHA_glu-VTA_DA synapses are enhanced by Dexamethasone (Dex) infusion (*n* cells group = 27, KS test, *D* = 0.37, *p* = 0.036). Example traces, scale bars: 25 pA, 25 ms. **b** Experimental timeline and schematic of in vivo local Dex infusion in the VTA. Below: Representative example of cannula placement with infusion site visualized by infusion of fluorescent beads (red). Scale bar: 100 μm. **c** Fat intake (kcal) overall increases in a 2 h limited fat access model after in vivo dex infusion (averages + SEMs, *n* animals control = 6, *n* animals Dex = 5, RM Two-way ANOVA. Main effect Dex, $F_{(1,9)}$ = 7.03, *p* = 0.03). **d** Timeline and schematic of in vivo optogenetic reduction of stress-potentiated LHA_glu-VTA_DA synapses. **e** Representative example images of coronal sections. Left: example of LHA CoChR (green) virus expression. Right: example of bilateral in vivo optogenetic fiber placement above the VTA with TH staining (purple) for dopamine

neurons. Scale bar: 100 μm. Anatomical reference point: f fornix. **f** Cumulative probability of LHA_glu-VTA opto-evoked AMPAR amplitudes for control, stress and stress + in vivo opto 1 Hz 30 min stimulation conditions. In vivo 1 Hz 30 min stimulation significantly decreases the AMPA amplitude, when comparing stress with and without stimulation (*n* cells control = 26, *n* cells stress = 22, *n* cells stress +1 Hz = 24, KS-test, *D* = 0.61, *p* = 0.0002). Scale bars: 25 pA, 25 ms. **g** Total fat intake (kcal) on PSD1 and PSD2 in 2 h access increased after stress in YFP control conditions, but not in CoChR conditions (averages + SEMs, n animals YFP-control = 30, *n* animals YFP-stress = 30, *n* animals CoChR-control = 21, *n* animals CoChR-stress = 19. RM Two-way ANOVA. Interaction stress X Virus type $F_{(1,96)}$ = 7.16, *p* = 0.009; YFP-Control vs YFP-Stress $F_{(1,58)}$ = 12.78, *p* = 0.001; CoChR-Control vs CoChR-Stress $F_{(1,38)}$ = 0.28, *p* = 0.60). All statistical tests were performed two-sided. *\**p* < 0.05, *\*\*p* < 0.01, *\*\*\*p* < 0.001. Source data are provided as a Source Data file.

that also in vivo application of the LFS protocol could longitudinally suppress LHA_glu-VTA synaptic strength in stressed mice.

We next assessed if LFS-mediated synaptic depression at LHA_glu-VTA afferents would prevent stress-induced eating. To this end, Vglut2-Cre mice were injected with a Cre-dependent CoChR in the LHA. As a control, Vglut2-Cre males were injected with a Cre-dependent YFP. All animals were implanted with an optic fiber above

the VTA and exposed to either the two-day control or stress protocol. All four groups were then given 30 min 1 Hz stimulation on PSD1 and subsequently given 2 h access to fat, and to chow for the remainder of the day. On PSD2 the mice were again given 2 h fat access and 22 h chow without any additional stimulation (Fig. 5d, e). As expected, the YFP group (effectively without LFS) showed a stress-driven increase in fat intake on PSD1 and 2. Importantly, in the CoChR group the fat

intake was decreased in the Stress+1 Hz group (effectively with LFS) on both PSD1 and 2 (Fig. 5g; Supplementary Fig. 7b). LFS did not modulate the effect of stress on chow intake observed in this experiment (Supplementary Fig. 7c). This implies that the stress-driven LHA$_{glu}$-VTA$_{DA}$ potentiation contributes specifically to stress-driven increased intake of fat.

## Discussion

Chronic stress in rodents is typically used to model depressive-like anhedonic reductions in sucrose preference (or of non-palatable chow)[29,38]. Instead, moderate stressors such as social subordination and immobilization stress, can increase intake of palatable food[27,39], but the neurobiological mechanisms for this behavior have remained largely unclear. Here we show that moderate social stress leads to increased intake of palatable food and we identify altered LHA control over the VTA dopamine system as a key neural substrate in this behavior.

We find that social stress directly excites LHA$_{glu}$ neurons projecting to the VTA, and that subsequently LHA$_{glu}$-VTA$_{DA}$ synapses are potentiated via a postsynaptic expression mechanism involving AMPA receptors. We evaluate this synaptic plasticity in a more integrated systems context and show that it results in enhanced excitatory, compared to inhibitory, output from LHA to VTA$_{DA}$, but not VTA$_{GABA}$ neurons. Moreover we implicate VTA$_{DA}$ neurons projecting to the mPFC (rather than to NAc medial shell) as one of the VTA neuronal subsets at which LHA$_{glu}$ synaptic plasticity occurs. We also show that stress-driven plasticity results in altered phasic LHA control over VTA dopaminergic output to the mPFC. Importantly, we demonstrate that stress-driven LHA$_{glu}$-VTA synaptic changes have consequences for palatable food intake. Mimicking both the acute and the synaptic plasticity effects of social stress on LHA$_{glu}$ inputs to VTA neurons increased fat intake in a limited access paradigm. Instead, weakening stress-potentiated LHA$_{glu}$-VTA$_{DA}$ synapses prevented enhanced fat intake.

Our findings suggest a previously unidentified role of LHA$_{glu}$ projections to the VTA in palatable food intake after stress. Previously LHA$_{GABA}$ cells and their projections to the VTA have been implicated in increasing food intake[20,21,23,31], while LHA$_{glu}$ neurons and those subsets projecting to the VTA, have instead rather been linked to encoding aversion and avoidance behavior[20,21,26,40]. Interestingly, recent studies performing single cell calcium recordings have found that subsets of LHA$_{glu}$ cells (including those projecting to the VTA) increase their activity to delivered sucrose reward[25,40,41]. However, these cells are preferentially active during the unexpected intake of unpleasant stimuli during which licking responses are curtailed[25]. Overall, this has resulted in conceptualizing LHA$_{glu}$ neurons as a braking system on food intake, of which the settings can be altered through internal state and/or experience. For instance the activity of LHA$_{glu}$ neurons in response to sucrose is decreased in mice rendered obese through exposure to high fat diet[41]. Our findings now elaborate on this idea. As found here and as previously described[20,21], LHA$_{glu}$ projections to the VTA are composed of both direct projections to VTA$_{DA}$ neurons and to, among other targets[22], VTA$_{GABA}$ neurons that can exert prominent feed forward inhibition onto those VTA$_{DA}$ neurons[21]. We hypothesize that the role played by LHA$_{glu}$ control over the VTA critically depends on the balance between its monosynaptic excitatory input and its disynaptic inhibitory input to VTA$_{DA}$ cells. Our data suggest that social stress re-equilibrates the LHA$_{glu}$-VTA circuit by shifting the balance more towards a direct excitatory one (LHA$_{glu}$-VTA$_{DA}$), resulting in enhanced drive to consume palatable fat.

Though the role of VTA$_{DA}$ neurons in food intake itself is complex[5], there is substantiating evidence implicating them and the potentiation of their glutamatergic synapses in (food) reward seeking. One study described that the exposure of mice to a high fat diet for 24 h resulted in behavioral priming for more rapid subsequent food

approach and consumption[42]. This effect coincided with enhanced glutamatergic input of unknown origin onto VTA$_{DA}$ cells and reducing this heightened glutamatergic drive via insulin application attenuated primed food approach and consumption[42]. Furthermore, infusing various orexigenic factors (e.g. ghrelin and orexin, the latter being expressed by subsets of VTA-projecting LHA$_{glu}$ neurons[25]) into the VTA leads to increased palatable food intake, and to both increased activity of VTA$_{DA}$ neurons and potentiation of their glutamatergic synapses of unknown origin[5,43–45]. Both ghrelin and orexin levels are elevated by stress[5] and might have a contributing role to the plasticity reported here. This could then be in addition to a possible role for GRs in this process. Indeed previous work identified GR-dependence of stress-driven potentiation of glutamatergic inputs of unknown origin onto VTA$_{DA}$ neurons[15,16]. We now describe that social stress leads to release of glucocorticoids and that a GR agonist can recapitulate potentiation of LHA$_{glu}$-VTA$_{DA}$ synapses and can induce increased palatable fat intake when infused in the VTA. These findings are also interesting in light of a recent study describing that a DA neuron specific knockout of GR abolished a conditioned place preference for high fat diet under a stressful condition[46]. Overall, we hypothesize that stress-driven LHA$_{glu}$-VTA$_{DA}$ plasticity occurs due to an interplay between initial stress-driven pathway activation, and subsequent corticosterone-GR and potentially ghrelin and orexin signaling.

What are the important downstream circuit elements beyond VTA$_{DA}$ neurons in mediating stress-driven food intake? In this study we provide evidence that stress plasticity affects at least LHA$_{glu}$ synapses onto mPFC-projecting VTA$_{DA}$ neurons. Moreover, we show that this plasticity alters phasic LHA control over dopamine output in the mPFC in vivo. Interestingly, the mPFC can be an important regulator of stress eating. It plays an important role in selecting behavioral strategies[11] and optogenetic stimulation of its dopamine 1 receptor-sensitive neuronal subset drives food intake[32]. We also observe that stimulation of mPFC$_{D1R}$ neurons can drive fat binge behavior in ad libitum fed mice. We did not observe consequences of LHA-VTA plasticity on dopamine release in the NAc medial shell. However, the NAc is composed of multiple subterritories with different responsivity to dopamine after salient stimuli[26]. Interestingly VTA$_{DA}$ terminals in the NAc medial but not the lateral shell are acutely activated by stressors[26]. In contrast to this acute response pattern, painful/stressful stimuli cause potentiation of glutamatergic synapses onto VTA$_{DA}$ neurons projecting to the NAc lateral, but not the medial shell[17]. Our findings therefore do not rule out that LHA$_{glu}$-VTA$_{DA}$ plasticity has consequences for hypothalamic control over dopamine release in other NAc subterritories than evaluated here. Further research will need to identify the exact downstream circuit elements through which altered LHA$_{glu}$-VTA$_{DA}$ synaptic strength exerts its precise role on food intake.

One limitation of the current study is that the ex vivo circuit mapping experiments were performed in the absence of TTX (and 4-AP) for polysynaptic block. Thus, the monosynaptic nature of the LHA-VTA recordings cannot be confirmed in this study. However, previous work has shown these connections to have strong monosynaptic components[20]. Moreover, it should be noted that multiple types of LHA glutamatergic neurons, including those that do or do not co-release orexin, project to the VTA[25]. Our current study has often co-activated these multiple pathways, and has not pinpointed which of these is/are affected by social stress. Future research will need to address this. One other notable point is that multiple VTA neuronal subsets receive LHA input. In our study we have assessed whether social stress differentially affects LHA input to VTA$_{DA}$ and VTA$_{GABA}$ neurons, as these often have opposite functions[5,6]. Notably however, there is more complexity in VTA neuronal subpopulations, including glutamatergic neurons and neurons releasing multiple neurotransmitters[6]. The VTA$_{glu}$ neurons, which are involved in defensive behaviors, receive a substantial input from LHA$_{glu}$ neurons[22]. Whether plasticity at LHA synapses onto VTA$_{glu}$ neurons is also

involved in stress eating behavior is an interesting possibility that needs to be addressed in future studies.

Overall our results contribute to further understanding of how specific stress-driven neural circuit changes, and the mechanisms underlying them, drive enhanced food reward intake. The neural circuits implicated here may be relevant also for other types of stress than social stress. Other stressors, like restraint, can also induce a preference for palatable food[39], though it remains unclear whether this occurs through LHA-VTA circuitry. Our findings may provide insights in the neural processes at play in individuals that cope with stress by overeating to a problematic extent. Interestingly, a recent human fMRI study found, in individuals with excess weight, that the resting state connectivity between the LHA and the midbrain was positively correlated to their stress response as well as to their tendency for emotional eating[47]. Our findings are in accordance with this, and provide a detailed view of the exact adaptations in this neural circuitry, and causally link them to stress eating responses. These findings can therefore be of importance in studying how stress contributes to problems in obesity and in psychiatric disorders characterized by (binge) eating.

## Methods

### Animals

In all experiments naïve adult male mice were used (20–35 g, >6 weeks). C57Bl6J (Jax #664), DR1D-Cre (Jax #28298), Pitx3-GFP mice were a kind gift from Meng Li (MRC Clinical Science Center)[48]. Pitx3-Cre mice were a kind gift from Marten Smidt (University of Amsterdam)[49]. Vglut2-Cre (Jax #28863) and VGAT-Cre (Jax #016962) animals were bred in house but originated from the Jackson Laboratory. The proven-breeder Swiss-CD1 mice (35–45 g, >12 weeks) were purchased from Janvier (France). All mice (except Swiss-CD1 mice) were group-housed (2–5 per cage) unless otherwise specified. Mice were housed in a 12:12 light/dark cycle (lights on at 07:00 a.m.) at $22 \pm 2\,°C$ (60–65% humidity). Unless otherwise specified, animals had access to ad libitum water and lab chow (3.61 kcal/g, CRM (E), 801730, Special Diets Services). Experiments were approved by the Animal Ethics Committee of Utrecht University and the Dutch Central Authority for Scientific Procedures on Animals (CCD), and were conducted in agreement with the Dutch law (Wet op de Dierproeven, 2014) and the European regulations (Guideline 86/609/EEC).

### Stereotactic surgery

For stereotactic surgery, all mice except cannula implanted mice were anesthetized with ketamine (75 mg/kg i.p.; Narketan, Vetoquinol) and dexmedetomidine (1 mg/kg i.p.; dexdomitor, Vetoquinol). The cannula implanted mice were anesthetised with isoflurane (Zoetis, UK, induction: 5%, maintenance: 1–2%). Lidocaine (0.1 ml; 10% in saline; B. Braun) was injected under the skin on the skull a few minutes before surgery. Eye ointment cream (CAF, Ceva Sante Animale B.V.) was applied before surgery. All mice were at least 6 weeks of age at the time of surgery. Animals were fixed on a stereotactic frame (UNO B.V. model 68U801 or 68U025) and kept on a heat pad during surgery (33 °C). When animals were implanted with either optic fibers or cannulas the skull surface was scratched with a scalpel and phosphoric acid (167-CE, ultra-Etch, ultradent, USA) was applied for 5 min to roughen the surface at the start of surgery. Injections were done using a 31 G metal needle (Coopers Needleworks) attached to a 10 μl Hamilton syringe (model 801RN) via flexible tubing (PE10, 0.28 mm ID, 0.61 mm OD, Portex). The Hamilton syringe was controlled by an automated pump (UNO B.V. -model 220). Injections were done bilaterally unless otherwise specified and volumes ranged between 250 and 300 nl per side, at an injection rate of 100 nl/min. The injection needle was retracted 100 μm, 9 min after the infusion and withdrawn completely 10 min after the infusion. The skin was sutured post-surgery (V926H, 6/0, VICRYL, Ethicon). Animals were then subcutaneously injected with the dexmedetomidine antagonist atipamezole (in case of ketamine, dexmedetomidine anaesthetic) (50 mg/kg; Atipam, Dechra), carprofen (5 mg/kg, Carporal) and 1 ml of saline and left to recover on a heat plate of 36 °C. Carprofen (0.025 mg/L) was provided in the drinking water for one week post-surgery. Animals were single-housed post-surgery and rejoined with their cage mates three days after surgery, the animals with additional fiber placements were single-housed for the remainder of the experiment.

**For Fiber photometric experiments.** Vglut2-Cre mice were injected unilaterally in the VTA with HSV-hEF1a-LS1L-GCaMP6s (Rachael Neve, MGH Harvard) or, for the control group, with HSV-hEF1a-LS1L-EYFP or -GFP (Rachael Neve, MGH Harvard) (−3.2 mm posterior to bregma, 1.60 mm lateral, −4.90 mm ventral under an angle of 15°). For the non-projection specific recording of LHA$_{glu}$ neurons, Vglut2-Cre mice were injected with in the LHA (−1.6 mm posterior to bregma, 1.80 mm lateral, −5.6 mm ventral from the skull, under an angle of 10°) with rAAV5-Syn-FLEX-GCAMP6S-WPRE-SV-40 ($3*10^{12}$ gc/ml). In all animals an optic fiber (400 μm, 0.50NA; FP400URT, Thorlabs) was implanted above the LHA (−1.6 mm posterior to bregma, 1.80 mm lateral, −5.3 mm ventral under an angle of 10°) inserted into a ceramic ferrule (bore 430–440 μm; MM-FER2006-4300, Precision Fiber Products). For extra grip four M1,2*2 mm screws (Fabory) were screwed into the skull 2*360° and a layer of adhesive luting cement (C&B Metabond; Parkell, Edgewood, NY, USA) was added. The cap securing the optic fiber was made with glass ionomer cement (GC Fuji PLUS capsules yellow, Henri Schein). Animals were allowed to recover for a minimum of four weeks before recordings were made.

**For ex vivo optogenetic-assisted electrophysiology experiments.** Pitx3-GFP mice were injected with AAV5-hSyn-hChR2(H134R)-mCherry-WPRE-PA ($2.85*10^{12}$ gc/ml; UNC Vector Core) in the Lateral Hypothalamus (LHA) (−1.3 mm posterior to bregma, 1.90 mm lateral, −5.4 mm ventral from the skull or −1.5 mm posterior to bregma, 1.70 mm lateral, −5.6 mm ventral from the skull under an angle of 10°). VGAT-Cre mice were injected as described for Pitx3-GFP, and an additional injection was placed in the VTA (−3.2 mm posterior to bregma, 1.60 mm lateral, −4.90 mm ventral, under an angle of 15°) with AAV5-EF1a-DIO-EYFP ($2*10^{12}$ gc/ml; UNC Vector Core) to visualize GABA neurons. Pitx3-Cre mice mice were injected as described for Pitx3-GFP, with two additional injections in the mPFC (1.9 mm anterior to bregma, 0.9 mm lateral, −2.2 mm ventral, under an angle of 10°) with retro-AAV-hsyn-DIO-mcherry ($2.8*10^{12}$ gc/ml; Addgene) +60x diluted (in PBS) red retrobeads (Lumafluor; for injection site visualization) and the NAc medial shell (1.7 mm posterior to bregma, 0.8 mm lateral, −4.9 mm ventral from the skull) with retro-AAV-hsyn-DIO-eGFP ($2.8*10^{12}$ gc/ml Addgene) +60x diluted (in PBS) green retrobeads (Lumafluor; for injection site visualization) to visualize dopamine neurons projecting to the mPFC and the NAc medial shell. All animals were allowed to recover for a minimum of four weeks before recordings were made.

**For array tomography experiments.** C56BL/6 J mice were injected with AAV5-hsyn-ChR(H134)-EYFP ($4.8*10^{12}$ gc/ml, UNC Vector Core) in the LHA (−1.3 mm posterior to bregma, 1.90 mm lateral, −5.4 mm ventral under and angle of 10°). Animals were allowed to recover for 6 weeks before experiment was started.

**For in vivo optogenetics combined with in vivo fiber photometric experiments.** Vglut2-Cre mice were unilaterally injected with AAV5-Syn-FLEX-CoChR-GFP ($2.9*10^{12}$ gc/ml, UNC Vector Core into the LHA (−1.3 mm posterior to bregma, 1.90 mm lateral, −5.4 mm ventral from the skull under and angle of 10°). An optic fiber (105 μm, 0.22 NA, FG105UCA, ThorLabs, Newton, NJ, USA) inserted in a ceramic stick ferrule (0.127–0.131 μm CFLC128-10, ThorLabs, Newton, NJ, USA) was implanted above the VTA (−3.2 mm posterior to bregma, 1.60 mm

lateral, −4.50 mm ventral under an angle of 15°). The dopamine sensor dLight was injected in the Nac medial shell or the PFC (AAVdj-hEf1a-dLight1.1-wpre-bGHp(A); 1*10¹² gc/ml; 400 nl) (NAc medial shell: 1.7 mm anterior to bregma; 0.8 mm Lateral; −4.9 ventral from the skull, PFC: 2.0 mm anterior to bregma; 0.6 mm lateral; −2.2 ventral from the skull). With the same anterior and lateral coordinates, an optic fiber (400 μm, 0.5 NA FP400URT, Thorlabs Newton, NJ, USA) inserted into a ceramic ferrule (bore 430–440 μm; MM-FER2006-4300, Precision Fiber Products) was implanted above respectively the NAc medial shell or the PFC (NAc: 4.75 mm ventral from the skull, PFC: 2.0 mm ventral from the skull). The fibers were secured as described above for the fiber photometric experiment. Animals were allowed to recover for six to eight weeks before experiments were performed.

**For in vivo optogenetic experiments.** Vglut2-Cre mice were injected with AAV5-Syn-FLEX-CoChR-GFP (2.2*10¹² gc/ml, UNC Vector Core) or AAV5-EF1a-DIO-EYFP (2.2*10¹² gc/ml; UNC Vector Core) into the LHA (−1.3 mm posterior to bregma, 1.90 mm lateral, −5.4 mm ventral under and angle of 10°). Optic fibers (105 μm, 0.22 NA, FG105UCA, ThorLabs, Newton, NJ, USA) inserted in a ceramic stick ferrule (0.127–0.131 μm bore, CFLC128-10, ThorLabs, Newton, NJ, USA) were implanted above the VTA (−3.2 mm posterior to bregma, 1.60 mm lateral, −4.50 mm ventral under an angle of 15°). For extra grip three M1,2*2 mm screws were screwed into the skull 2*360° and a layer of adhesive luting cement (C&B Metabond; Parkell, Edgewood, NY, USA) was added. The cap securing the optic fiber was made with glass ionomer cement (GC Fuji PLUS capsules yellow, Henri Schein). Animals were allowed to recover for a minimum of eight weeks before experiments were done.

**For in vivo cannula infusion experiments.** C56BL/6 J mice were bilaterally implanted with a guide cannula (4 mm; C315GMN/SPC; Bilaney) above the VTA (−3.2 mm posterior to bregma, 1.60 mm lateral, −3.7 mm ventral under an angle of 15°). The cannulas were secured as described above for the ferule implantations. Animals were allowed to recover for at least one week before experiments were done.

**For in vivo chemogenetic stimulation experiments.** DRD1-cre mice were injected with pAAV5-hSyn-DIO-mCherry (3*10¹² gc/ml Addgene) or rAAV5-hSyn-DIO-hM3D(Gq)-mCherry (3*10¹² gc/ml Addgene) in the mPFC (1.9 mm anterior to bregma, 0.6 mm lateral, −2.0 mm ventral from the skull, under an angle of 10°). Animals were allowed to recover for five weeks before experiments were done.

### Behavioral paradigms
All behavioral manipulations and tests were performed during the light phase.

**Two-day social stress paradigm.** The social subordination stress protocol used was designed according to a resident intruder paradigm. The resident, a Swiss-CD1 male previously used for breeding, was housed in a large Makrolon cage (type IV, Tecniplast, Italy), an intruder; an experimental male mouse, was placed in this cage. Fighting time was tracked live and animals were allowed to fight for 20 s, twice a day for two days, one fighting session in the morning (8.30–11.00 a.m.) and one in the afternoon (16.00–18.30 p.m.). For the remainder of the day, animals were separated by a perforated transparent splitter, allowing sensory, but no physical interaction. The control animals were co-housed with a non-familiar C56BL/6 J male mouse for two days, preventing physical interaction.

**Elevated plus maze and light-dark box.** The elevated plus maze (EPM) was performed using a standard apparatus (arm length: 65 cm × 65 cm, height: 65 cm with 15.0 cm high walls delimiting the enclosed arms. Light intensity in the neutral zone of the maze was 30 Lux. Mice were placed in the EPM in the neutral zone facing in the direction of an open arm. Mice were allowed to freely explore the EPM for 10 min on post stress day 1 (PSD1). Head dips were visually quantified by an experimenter blind to the experimental group. A head dip was counted when the mouse fully extended the head outside an open arm. The Light-Dark Box (LD Box) comprised of 2/3 light area (29 × 21 cm) and 1/3 dark area (14.5 × 21 cm) connected by a 5 × 5 cm opening. The height of the box was 21 cm. Light intensity was set at 80 Lux in the light compartment. Mice were placed in the dark compartment at the beginning of the test. Mice were allowed to freely explore the LD box for 10 min on PSD1. EPM and LD box were cleaned with Anistel (1:200) in between animals. Data collection and analysis for both EPM and LD box was done with Ethovision video tracking (version 9; Noldus).

**Ad libitum palatable choice diet.** The High Fat High Sugar (HFHS) diet consisted of Lard (Blanc de Boeuf, Vandemoortele, 9.0 kcal/g), 10% sucrose solution (granulated sugar weight/volume, 0.387 kcal/g), regular chow (3.61 kcal/g) and water. Habituation to fat and sugar was done in the home cage one week before the start of the stress for one day. Baseline chow intake was monitored prior to and during the social stress. Two days of social stress were done as described above. Animals were kept in either control or stress conditions without allowing physical interaction for PSD1-2 while food intake was measured daily. HFHS diet was provided after the last fight on stress day 2. The HFHS diet intake was quantified per 24 h starting in the morning of PSD1. For sugar water intake a drip-factor was calculated per day, based on sugar water loss over 3 days without animal interaction and subtracted from the consumed amount per day.

**Limited access binge model.** In the limited access paradigm, animals were exposed to fat (i.e. lard; Blanc de Boeuf, Vandermoortele, 9.0 kcal/g) for 2 h a day in a clean cage (Makrolon cage type IV, Tecniplast, Italy), drinking water was provided. Animals were habituated for 1 h to the limited access cage and fat, four days before the start of the social stress. Animals were subjected to stress as described above and on PDS1 and 2, fat intake was measured over a 2-hour fat access period, starting between 10 a.m. and 11 a.m.. Unlimited chow was provided the remaining 22 h of the day and intake measured.

**Corticosterone measurements.** To obtain corticosterone measures animals were anaesthetized with isoflurane (Zoetis, UK) and decapitated. Trunk blood was collected and stored on ice in heparin containing tubes (Sarstedt, The Netherlands). Blood samples were centrifuged for 10 min, plasma was stored at −20 °C until radio-immunoassay (MP Biomedicals, The Netherlands; sensitivity 3 ng/ml). To determine corticosterone levels after acute 20 s of fighting, Pitx3-GFP animals were introduced in the home cage of Swiss-CD1 mice and allowed to fight for 20 s. After which they were separated by a perforated splitter as described in the 2-day social stress protocol, 30 min later blood samples were collected. For the control condition, animals were placed in a clean cohousing cage for 30 min, apart from the duration, identical to the control condition described in the 2-day social stress protocol, and blood samples were collected. All samples were collected between 11 a.m. and 1 p.m.

### In vivo fiber photometric measurements
Fiber photometry experiments were performed using a Doric set up. The data acquisition unit and fiber photometry console (FPC, Doric) were connected to the dual channel programmable LED driver (400 mA/V in analogue mode LEDRVR_2CH_1A, Doric), which in turn controlled the maximum power (950 mA) output to the light sources: a blue LED for excitation of calcium dependent GCaMP6s (440–490 nm, LEDC1-B_FC, Doric). The LEDs were connected with a 200 μm core cable (MFP_200/240/LWMJ 0.221m_FCM-FCM, or T0.40, Doric) to their respective input ports of a Doric minicube (Doric FMC5_AE(405)

_AF(420450)_E1(460-490)_F(500-550)_S). A 400 μm core patch cord (MFP_400/460/900-0.48_3m_FCM-MF2.5) was attached to the sample port of the minicube and a 2.5 mm ceramic ferrule sleeve (SM-CS1140S, Precision Fiber Products) was connecting the fiber tip with the animal's fiber implant during the experiments. Individual LED power was set at 120–160 μW in the brain.

The emitted fluorescence, went through the same 400 μm patch cable back to the minicube, where it was split by two dichromatic mirrors (420–450 nm and 460–490 nm), and was sent via 600 μm core cables (MFP_600/630/LWMJ-0.48_1m_FCM-FCM) to photoreceivers (Newport 2151 with lensed FC adapter). Photon-to-electron conversion took place in AC low mode (gain $2 \times 10^{10}$ V/A, Bandwidth 30-750 Hz) and signal was sent back to the fiber photometry console for demodulation. Recordings took place in lock-in mode, using 208 Hz as reference frequencies for blue (465 nm) light channel. Data were acquired at 12 kilosamples/s and decimated 50 times using Doric Neuroscience Studio. Finally, dF/F0 was calculated in Doric Neuroscience Studio software as 100* (F-F0)/F0 where F0 was defined as the running average of a 60 s time-window. We then aligned the onset of each behavior with the photometric df/F signal using custom-made python scripts.

For acute social stress, juvenile interaction and locomotion experiments the photometric system was coupled to a behavior tracking camera (BTC_USB3.0_CO, Doric), to be able to record behavioral videos synchronised to emitted fluorescence signal. For the foot shock experiment, the med-associates input/output smart control box (DIG-716B, Med-associates) was connected to an analogue input port on the fiber photometry console and sent TTL pulses when a shock was being delivered, recorded transients were time-locked to these TTL pulses i.e. shock onset.

All mice were handled one week prior to the experiment and habituated to the cables for 3 days prior to the experiment (30 min/day). For experiments assessing responsivity to stressors, all behavioral measurements took place on the same day, in the same order, starting with locomotor behavior followed by juvenile interaction, acute social stress and 2–4 h after the acute social stress, foot shock. For experiments assessing fat intake responses, a separate batch of experimental animals were utilized. These animals were additionally habituated to an operant chamber with fat, five days prior to the baseline recording.

**Fiber photometry acute social stress.** Experimental animals were introduced into the cage of the Swiss-CD1 aggressor and photometric signal and video was recorded in synchrony while animals fought. Videos were manually scored and fights were selected according to the following criteria: duration of at least 6 s from 1st bite or touch, no fight taking place in the 3 s prior to each fight onset and 6 s after each fight offset. Calcium dependent transients were time-locked to fight onset and fight offset of selected fights and averaged for at least 5 fights per animal.

Fights were also analysed with an automated pipeline, using DeepLabCut (version 2.0.6)[50] for pose estimation, SiMBA[51] for machine learning (behavioral) classification and custom-made python scripts for time-locking photometric signal to behavior as follows: On DeepLabCut we labelled 16 total body parts on two mice (8 each) across 437 frames taken from 8 videos. 95% of them was used for training. We used a ResNet-50 based neural network with default parameters. The network was trained and refined for 5 rounds in total (200.000–300.000 iterations each). In final body part estimation analysis we had 1.96 pixels training error and 9.1 pixels test error (at p-cut-off 0.3). This network was used to analyze all fight videos. Then SimBA was used, with a configuration of 2 animals and 14 body parts. For annotation of fight behavior we used the manual scoring, by converting manual timestamps to frame-by-frame annotation, from 6 videos (5 of which were also part of the DeepLabCut training).

A Random Forest (RF) classifier was then trained and evaluated (number RF estimators = 2000, training test fraction 0.2, default hyperparameters). We used a discrimination threshold of 0.5 and minimum fight bout length of 1000 ms. Finally this classifier was used to analyze all videos. The predicted fighting bouts were further analyzed using custom Python scripts: fight predictions <1 s were removed and predictions of <4 s in between them were merged. Then final fight bouts were selected on the following criteria: 3 s fight duration, 3 s of no aggressive occurrence as baseline, as well as after the offset of the fight. Timestamps for fight onset were extracted and time-locked with photometric signal. Our pipeline has a True Positive Rate TPR = 0.72 + −0.04 (Sensitivity), Positive Predictive Value PPV = 0.84 + −0.04 (precision) and F1-score = 0.76 + −0.03 (harmonic mean of sensitivity and precision). One animal in the GFP group was excluded as it did not reach the criteria for minimum detected fight duration.

**Fiber photometry acute foot shocks.** Experimental animals were placed in operant chamber modified for foot shock delivery (29.53 × 24.84 × 18.67 cm, Med Associates) and were given 5 min to habituate to the chamber. A total of 20 foot shocks of 1 s and 0.75 mA were delivered with an interval of 90 s. Calcium transients were time-locked to shock onset and averaged per animal.

**Fiber photometry juvenile interaction.** Animals were introduced to a juvenile mouse (male C56BL/6 J, 8–10 days old) in a clean cage (Makrolon type II, Tecniplast, Italy). After 10 min of habituation to the environment and juvenile mouse, photometric signal and video were recorded in synchrony. The recording was stopped when animals had at least 5 interactions with a minimum of 6 s. Interactions between the experimental and juvenile mouse were manually scored. Interaction onset was defined as the moment when the nose of the experimental mouse touched the juvenile mouse. Interaction was then labelled as such if the experimental animal continued to sniff the juvenile mouse. The interactions selected for further analysis lasted more than 6 s, while no interaction occurred in the 3 s prior to the onset and 6 s after the offset. Calcium transients were then time-locked to the selected interaction onset and averaged for at least 5 interactions per animal.

**Fiber photometry locomotor activity.** Experimental mice were placed in a clean cage (Makrolon type III, Tecniplast, Italy). After 2 min of cage habituation, a 15-minute photometric recording was initiated in synchrony with video recording. For body part tracking we used DeepLabCut (version 2.0.6)[50]. Specifically, seven body parts were labeled across 497 frames taken from 11 videos. 95% of them was used for training. We used a ResNet-50 based neural network with default parameters. The network was trained and refined for three rounds in total (250.000–350.000 iterations each). In final body part estimation analysis we had 1.79 pixels training error and 8.8 pixels test error (at p-cut-off 0.1). We then used tools from SiMBA[51] to perform the pixel/mm conversion and outlier correction. Velocity was calculated per frame as the movement (mm) of the central body part / framerate (ms). Output was further analyzed using python scripts made in house as follows: high velocity frames were defined using velocity threshold of at least 0.15 m/s. A high velocity bout was then defined as a period containing >2 high velocity frames with max time between them of rounded 1 s. Timestamps of bout onset and offset were then time-locked to Calcium transients. Two animals were excluded from analysis as they did not reach this high speed.

**Fiber photometry fat intake.** For this experiment, an operant chamber (30.5 cm × 24.1 cm × 33 cm, MedPC Associates) with a curved rear wall and a grid floor was used. A total of 5 square cue holes (2.54 cm × 2.2 cm × 2.54 cm) were set in the curved wall, equipped with an infrared beam. A small block of fat (Blanc de Boeuf, Vandermoortele, 9.0 kcal/g) was placed in one cue hole and a transparent plexiglass floor was

placed above the grid for easier access to the fat. Before photometric measurements were taken, all mice were habituated to the operant chamber for 1 h with free access to the fat. At least 3 days later, animals were subjected to 1–2 baseline photometric recording sessions after which animals were subjected to either 2 days control or social stress. On PSD1/2, animal were recorded again in one or two sessions. For the control stress animals, a new baseline was recorded a week later followed by social stress and a post stress recording session. During each session, mice had unlimited access to fat for 1 h. Access to fat was measured in nose pokes, registered by the MedPC software through the infrared beam-breaks. For each nose poke a TTL pulse was generated and recorded by the Doric set up. The calcium transients were then time-locked to each TTL pulse for at least 15 nose pokes for fat and averaged per animal.

**In vivo optogenetics combined with in vivo fiber photometric experiments.** These experiments were performed with the fiber photometric system as described above, with the only differences being the excitation LED (CLED_465, Doric) and the patch cord (MFP_400/430/1100-0.57_3m_FCM-MF2.5_LAF). Also a 473 nm diode-pumped solid state (DPSS) laser (Changchun New Industries Optoelectronics) was connected on the Med-associates input/output smart control box, and was controlled by MedPC software to deliver the stimulation protocol. The light was guided from the laser, through optogenetic cables made in house (105 μm, FG105LCA, Thorlabs) to the optogenetic implant on the mouse's head. The same input/output box was also connected with Analog Input channel of the fiber photometry console, to deliver TTL pulses upon pulse onset. This allowed for time-locking of signal on burst onset. dLight excitation light was set at 89–155 μW in the brain, and stimulation intensity was 15 mW in the brain.

Vglut2-Cre male mice were handled two weeks before recordings and habituated to optogenetic and fiber photometric cables 1–2 times a day, every day, a week prior to the recordings. Once animals were habituated, two baseline measurements were done which were 3–6 days apart. For baseline measurements, $LHA_{glu}$ –VTA terminals were optogenetically stimulated while measuring emitted fluorescence by the dLight vector in the NAc medial shell or the mPFC. For the NAc medial shell the stimulation protocol consisted of continuous 10 Hz (5 s duration, 10 ms pulse width) and 20 Hz (2.5 s duration, 10 ms pulse width). For the mPFC the stimulation protocol consisted of 10 s stimulation trains of alternating 200 ms of stimulation and 300 ms of no stimulation. During the 200 ms stimulation periods bursts of 20, 33 or 50 Hz (in different blocks) were delivered (10 ms pulse width). On the next day all animals were subjected to two days of social stress as previously described. On post stress day 1, animals were again subjected to optogenetic stimulation trains of 20, 33 and 50 Hz, time-locking the emitted fluorescence to stimulation train onset using custom-made Python scripts. For quantification of the results, maximum difference amplitudes were determined during stimulation opposed to a 5 s pre stimulation baseline. This was done for both pre stress baseline days as well as for post stress separately. For the mPFC pre-stress baseline day 1 and 2 were averaged for each stimulation frequency. All peak values were hereafter normalized towards the average peak of pre stress day 1 and 2 baseline during 20 Hz stimulation.

## Patch-clamp electrophysiology

Animals were anesthetized with isoflurane (Zoetis, UK) between 8.30 a.m. and 10 a.m. and then rapidly decapitated. Horizontal brain slices of 250 μm were cut on a vibratome (1200 VTs, Leica, Rijswijk, The Netherlands) in ice-cold carbogenated (95% $O_2$, 5% $CO_2$) cutting solution, containing (in mM) choline chloride 92; ascorbic acid 10; $CaCl_2$ 0.5; glucose 25; HEPES 20; KCl 2.5; N-Acetyl L Cysteine 3.1; $NaHCO_3$ 25; $NaH_2PO_4$ 1.2; NMDG 29; $MgCl_2$ 7; sodium pyruvate 3;

Thiourea 2. Slices were transferred for 5 min to warmed solution (34 °C) of identical composition, before they were stored at room temperature in carbogenated incubation medium containing (in mM) ascorbic acid 3; $CaCl_2$ 2; glucose 25; HEPES 20; KCl 2.5; NaCl 92; $NaHCO_3$ 20; $NaH_2PO_4$ 1.2; NMDG 29; $MgCl_2$ 2; sodium pyruvate 3 and Thiourea 2. During recordings, slices were immersed in artificial cerebrospinal fluid (ACSF) containing (in mM) $CaCl_2$ 2.5; glucose 11; HEPES 5; KCl 2.5; NaCl 124; $NaHCO_3$ 26; $NaH_2PO_4$ 1; $MgCl_2$ 1.3 and were continuously superfused at a flow rate of 2.5 ml min −1 at 32 °C.

Neurons were patch-clamped using borosilicate glass pipettes (2.7–4 MΩ; glass capillaries, GC150-10, Harvard apparatus, UK), under a TH4-200 Olympus microscope (Olympus, France). For voltage or current clamp recordings, signal was amplified, low-pass filtered at 2.9 kHz with a 4-pole Bessel filter, and digitized at 20 kHz with an EPC10 dual patch-clamp USB amplifier (HEKA Elektronik GmbH). Data were acquired using PatchMaster v2x90.2software. Access resistance was continuously monitored with a − 4 mV step delivered at 0.1 Hz. Experiments were discarded if the access resistance increased by more than 20% during the recording. Light pulses (470 nm, 1–10 ms; 2–44 mW) were delivered with a 300Ultra, CoolLed, UK illumination system. All electrophysiological measures are recorded with a 10 s inter sweep interval (0.1 Hz) and all data points are shown as an average of 10–20 sweeps, unless otherwise specified.

**AMPAR-NMDAR ratios.** Synaptic LHA optogenetically evoked AMPAR and NMDAR currents were measured in response to a single light pulse. Recordings were made in Voltage clamp with a cesium methanesulfonate-based internal medium containing (in mM) cesium methanesulfonate 139, CsCl 5, HEPES 10, EGTA 0.2, creatine phosphate 10; Na2ATP 4; Na3GTP 0.3; $MgCl_2$ 2. Under these conditions, it was possible to detect inward glutamatergic currents (at −65 mV) and NMDAR (at 40 mV at 100 ms) pharmacologically isolated (by 100 μM picrotoxin; $GABA_A$ receptor antagonist). When clamping neurons at −65 mV, the inward current was fully blocked by AMPA/Kainate receptor antagonist CNQX (10 μM). The amplitude of the glutamatergic current measured at 100 ms after opto-stimulation, was blocked by NMDAR antagonist APV (50 μM). For a second measure of AMPAR/NMDAR ratio both AMPAR and NMDAR currents were recorded at +40 mV. Subsequently the NMDAR-mediated component was blocked by APV. The NMDAR current was then determined by subtracting the remaining AMPAR-mediated current from the total current at +40 mV, and taking the ratio between the peak values of the AMPAR- and the NMDAR-current.

**AMPAR rectification.** LHA AMPAR currents were recorded at −60 mV, at 0 mV and at +40 mV while blocking $GABA_A$Rs and NMDARs (by picrotoxin 100 μm and APV 50 μM, respectively) in the same buffers as described above. For rectification plots, the synaptic currents at 0 mV and +40 mV for a cell were multiplied by −1 and divided by the total current at −60 mV. The synaptic current at −60 mV was then set at −1. The AMPAR rectification index was calculated by dividing the peak of the synaptic response at −60 mV (absolute current) by the peak of the synaptic response at +40 mV, and dividing this value by 1.5.

**LHA-VTA connectivity experiments and $GABA_A$R-AMPAR ratios.** Recordings were made in voltage clamp in a cesium methane sulfonate based internal medium as described above. Under these conditions, it was possible to detect inward glutamatergic currents (at −65 mV) and outward GABAergic currents (at +0 mV). When clamping neurons at 0 mV, outward currents were fully blocked by $GABA_A$ receptor antagonist bicuculline (20 μM). $GABA_A$R-AMPAR ratios were calculated by dividing the peak synaptic response at −65 mV (absolute current) by the peak response at 0 mV. Cell type specific connectivity was

determined based on whether a neuron showed an LHA opto-evoked synaptic response bigger than 5 pA.

**Spontaneous excitatory and inhibitory postsynaptic currents.** Recordings were made in voltage-clamp in a cesium methane sulfonate based internal medium as described above. Spontaneous excitatory postsynaptic currents (sEPSCs) were recorded at −65 mV, while spontaneous inhibitory postsynaptic currents (sIPSCs) were recorded at 0 mV. Recordings of at least 10 min were included. Spontaneous events were subsequently detected and analyzed in Mini-Analysis (Synaptosoft). Histograms of event frequencies (bin size 0.1 Hz) and amplitudes (bin size 1 pA) were calculated. Average instantaneous frequency and amplitudes were derived from the resultant log-normally distributed histograms in Igor Pro using f(x) = A*exp(−0.5*(((Ln(x)·mu))/sigma)^2)/(sigma*x*sqrt(2*pi)) as an initial fitting function to obtain distribution parameters. From there the mean (m) of the underlying normal distribution was assessed by m = exp(μ+sigma²/2).

**Intrinsic excitability and cell-attached recordings.** Recordings were made in current clamp in a potassium gluconate based internal containing (in mM), Potassium Gluconate 139; HEPES 10; EGTA 0.2; creatine phosphate 10; KCl 5; Na2ATP 4; Na3GTP 0.3; MgCl$_2$ 2. To assess the membrane resistance, firing pattern and voltage sag neurons were subjected to 12 subsequent current steps of 800 ms length, starting from −250 pA, with a 50 pA increasing increment/step. For cell-attached, recordings were made in voltage clamp with clamp potentials continuously set to nullify currents through the recording amplifier head stage. Regular ACSF (composition as described for recording procedures) was used both as an extracellular medium and as a pipette medium. Cells were patched and were kept sealed (300 MOhm – 1 GOhm). Cells were kept in this configuration for 10 min prior to the onset of recordings (10 min) of the occurrence and frequency of action potentials.

**Paired-pulse ratio (PPR).** PPR experiments for LHA-driven AMPAR- and GABA$_A$R-mediated currents were performed with the same cesium methanesulfonate-based internal medium described above. AMPAR and GABA$_A$R mediated currents were measured as described for the AMPAR-GABA$_A$R ratio. However, now in response to opto-stimulation of LHA terminals using two pulses with a 100 ms inter-pulse interval and measuring the resulting peak amplitude of the two synaptic responses. The PPR was calculated by dividing the amplitude of the synaptic response to the 2nd pulse by the amplitude of the synaptic response to the 1st pulse.

**Ex vivo long term depression (LTD).** Recordings were made in voltage clamp in an internal medium containing (in mM), Cesium Chloride 139, HEPES 10, EGTA 0.4, creatine phosphate 10; NaCl 5; Na2ATP 4; Na3GTP 0.3; MgCl$_2$ 2. Initially, a stable baseline was obtained for LHA optogenetically evoked inward AMPAR current at −65 mV for a minimum of 5 min in the presence of a GABA$_A$R blocker (picrotoxin 100 μM). To induce LTD a low frequency stimulation (LFS) of 1 Hz was delivered for 10 min while clamping the neuron at −65 mV. AMPAR currents were subsequently measured for 30 min. During baseline and 30 min post LFS; LHA stimulation was as described for PPR recordings. The data post LTD induction was normalized to baseline and all data points were binned per minute. For quantification of the LTD the binned normalized data points of the last 10 min post LTD were averaged.

**Ex vivo validation of dexamethasone infusion.** Pitx3-GFP animals were injected with ChR2 in the LHA and sliced as described above. Thereafter slices were incubated in 100 μM dexamethasone (or in 0.008% DMSO alone for vehicle control) in the incubation ACSF for

30 min. Recordings were then made in voltage clamp in a cesium chloride-based internal medium as described above. LHA optogenetically-evoked inward AMPAR currents were measured at −65 mV in the presence of a GABA$_A$R blocker (picrotoxin 100 μM), or at −60, 0 and +40 mV for AMPAR rectification as described above.

**Ex vivo validation of in vivo LFS.** Vglut2-Cre animals injected with Cre-dependent CoChR (AAV5-Syn-FLEX-CoChR-GFP) used for control condition in the in vivo opto-LFS + limited access food intake (Fig. 5g) were redistributed into 3 groups. At the start, animals were subjected to control stress or 2 days of social stress, the stress group was then subdivided in 2 groups receiving LFS or mock LFS, while the control group received no LFS. The first stress group received a mock LFS (i.e. 30 min attached to in vivo stimulation set up without light) and the second stress group received 30 min of 1 Hz in vivo LFS stimulation. On PSD1 after receiving no LFS, mock LFS or LFS animals were anaesthetised with pentobarbital (100 mg/animal) and transcardially perfused for 4 min with ice-cold cutting solution of which the composition is identical to the choline-based solution described above. Brains were further processed as described for the other electrophysiological recordings. Putative dopamine neurons were patched based on shape and size. Recordings were made in voltage clamp in a cesium methanesulfonate based internal solution identical to the medium described above. Peak AMPAR amplitude was measured at −65 mV.

**Drugs for patch-clamp experiments.** All drugs were obtained from Sigma (UK), Abcam (UK) and Tocris (UK). With the exception of picrotoxin (DMSO, 0.01% final bath concentration) and dexamethasone (DMSO, 0.008% final bath concentration), all drugs were dissolved in purified water. Drugs were aliquoted at a concentration of 1/1000th of the final bath concentration and stored at −20 °C. Upon use they were thawed and quickly dissolved in the recording medium.

### Array tomography

**Tissue processing.** C57Bl6J mice were anesthetized with pentobarbital (75–100 mg/kg) and perfused transcardially with 4% paraformaldehyde in 0.1 M phosphate buffer (PB) (pH 7.4). Brains were removed and post-fixed overnight at 4 °C, then equilibrated in 30% sucrose in phosphate-buffered saline (PBS) for 48 h. Coronal sections containing the Ventral Tegmental Area were cut into 300-μm-thick sections in a vibratome. Then, sections were further processed for array tomography as before[52]. Briefly, tissue sections were dehydrated in series of alcohol up to 70% ethanol and then infiltrated overnight at 4 °C in LRWhite resin (medium grade, Electron Microcopy Sciences, USA; #14380). Next, sections were flat embedded using a glass slide and ACLAR plastic (Electron Microcopy Sciences, USA; #50425-10) and polymerized for 24 h at 55 °C. After embedding, the region of interest was excised from tissue sections and glued onto EMBed 812 blocks for ultrasectioning. Adhesive cement diluted in xylene was applied to the top and bottom of the block face to allow the collection of serial sections. Series of 25 or more 100 nm-thick sections were cut in ribbons using Jumbo Histo Diamond Knife (Diatome, Switzerland) and an ultramicrotome (Leica, France). The ribbons were mounted onto glass coverslips coated with 0.1% gelatin and 0.01% chromium potassium sulfate, air dried, placed on a hot plate (60 °C) for 30 min, and then stored at room temperature until immunolabeling.

**Immunofluorescence.** All the antibodies used have been previously characterized and validated for array tomography[53]. These include: primary antisera anti-Vglut2 (guinea pig, 1:1000; Millipore, Germany; AB2251), and monoclonal antibodies anti-synapsin 1 (rabbit, 1:400; Cell Signaling Technology, USA; #5297) (clone D12G5), anti-GluA1 (mouse, 1:200; Millipore, Germany; MAB2263) (clone RH95), and anti-tyrosine hydroxylase (mouse, 1:400; Millipore, Germany;

MAB318) (clone LNC1). Antibodies from different hosts were applied together in different rounds of immunolabeling followed by antibody elution. DAPI staining was included in each round for alignment of serial-images. For immunolabeling, sections were encircled with a hydrophobic barrier pen (ImmEdge, Vector Labs, USA; #H-4000). Primary antibodies were diluted together in blocking solution (0.05% Tween; 0.1% bovine serum albumin in Tris-buffered saline; pH 7.6) and incubated with sections for 2 h. Subsequently, sections were thoroughly rinsed with PBS three times for 10 min each with a transfer pipette. Appropriate combinations of fluorescence-conjugated secondary antisera raised in donkey were used (Alexa 488, Alexa 647, and CY3, 1:200; Jackson ImmunoResearch, USA). Sections were incubated with the secondary antisera for 25 min and then rinsed with PBS. The coverslips with sections were mounted on a glass slide using the SlowFade Gold Antifade Mountant with DAPI (Life Technologies, USA; S36939) and immediately imaged. Elution of antibodies was done with 0.02% SDS and 0.2 M NaOH in distilled $H_2O$ for 20 min. After two 10 min washes with distilled $H_2O$, coverslips were air-dried and placed on a hot plate (60 °C) for 30 min. Negative controls omitting primary antisera were run to corroborate the complete elution of primary antibodies.

**Image processing and quantitative analysis.** Serial images were acquired on a Leica DM6000 fluorescence microscope with a 63X NA 1.4 Plan Apochromat oil objective, a CoolSNAP EZ camera, and Metamorph software (Molecular Devices, USA). Multiple channels of serial images were aligned and converted into stacks with Fiji software and the StackReg and MultiStackReg plugins, using DAPI staining as a reference channel[52]. We analyzed two stacks of at least 25 images per animal to obtain a mean value of density of synaptic boutons per $\mu m^3$ of tissue in each mouse. In the quantitative analysis we used a sampling mask of at least 90 μm X 90 μm. Images were converted to binary images using the threshold function and in some cases manually adjusted to exclude non-specific background. To avoid underdetection of adjacent objects located in close proximity to each other, we used the "dilate" function that introduces a mask expansion of <0.2 μm in one of the objects. In these analyses, we used this resulting dataset to quantify the density of labeled overlapping voxels[52]. For the statistical analysis, array tomography data obtained from two stacks per animal were averaged to generate a mean density value per mouse. Data were collected and analyzed blind to the experimental groups.

## In vivo circuit manipulation and food intake experiments
**In vivo local dexamethasone infusion and limited access model.** C56BL/6 J male mice were handled and habituated to the infusion tubing with an injector dummy (0.1 mm projection; C315FDMN/SPC; Bilaney) for three days, 2–3 times a day. On day 2 after surgery animals were given a bit of fat in the home cage to prevent neophobia. Five days after surgery animals were habituated to the limited access to fat paradigm (identical to the description above) for two consecutive days. On day 10 after surgery (experimental day 1) the infusion experiment took place. The injector needles (1 mm projection; C315IMN/Spc; Bilaney) glued to tubing (PE10, 0.28 mm ID, 0.61 mm OD, Portex) were bilaterally inserted in the cannulas. The tubing was attached to 10 μl Hamilton syringe (model 801RN). The Hamilton syringe was controlled by an automated pump (UNO B.V. -model 220). While the animals were in their home cage, 1 μg dexamethasone was infused in a volume of 300 nl (10% DMSO, 90% PBS) in each hemisphere. The injector was left in place for an additional 5–10 min after infusion. 30 min after infusion animals were placed in the limited access paradigm for two hours and chow intake was measured for the 22 h following the 2 h fat exposure. On experimental day 2, animals were placed in the limited access paradigm without additional infusion.

**In vivo optogenetic LFS and limited access model.** Vglut2-Cre male mice were handled and habituated to the cables of the in vivo optogenetic set-up while in their home cage (>5 weeks after surgery) for 3 weeks. The animals were attached to cables for a few minutes every other day. After the 3 week habituation to the cables animals were familiarized with the limited access cage for 2 h where they had access to fat. The limited access cage with fat was as described above.

Three days later, animals were stressed with the 2-day social stress paradigm, as described above. On post stress day 1 animals were placed in the in vivo optogenetic stimulation set up. This set-up consisted of a clean Makrolon cage (type II, Tecniplast, Italy) in which the mice could move around freely while the cables were connected to the implants. Cables were in turn connected to a commutator (rotary joint; Doric, Québec, Canada) via an FC/PC adapter to allow unrestricted movement. A second patch cable, with a FC/PC connector at both ends (Doric, Québec, Canada), was connected to the commutator and then connected to a 473 nm diode-pumped solid state (DPSS) laser (Changchun New Industries Optoelectronics). Lasers were attached and commanded by Med-PC IV (Med Associates). Animals were stimulated bilaterally with the 473 nm laser for 30 min at a frequency of 1 Hz, 10 ms pulse length, 8-10 mW light intensity. After the end of this 30 min stimulation period, mice were disconnected from their cables and placed in the limited access cage with fat as described earlier. The remaining 22 h they had access to chow in their home-cage and their chow intake was monitored. Note that, identical to limited access model described earlier, animals continued to be cohoused with another C57Bl6J background male (control condition) or Swiss-CD1 aggressor (Stress condition) during the post stress days. On post stress day 2, fat intake was measured in limited access paradigm and chow intake for the remainder of the day as for post stress day 1, however without additional optogenetic stimulation.

**In vivo optogenetic HFS and limited access model.** Vglut2-Cre mice prepared as in the in vivo optogenetic LFS and limited access model were given a baseline fat exposure in the limited access binge model as described above. They were then exposed to 10 min of high frequency stimulation (HFS; 20 Hz, 5 s ON, 10 s OFF) per day (9–11 am) on two days. On Post HFS day 1 their chow intake and their fat intake in another cage were measured (animals were at this stage no longer attached to the optic fiber).

**In vivo chemogenetic stimulation and limited access model.** DRD1-Cre mice were handled for three days and habituated to restraint and mock IP injections daily. Animals were given fat access in the limited access binge model as described above for 2 h for 2 days. On the third day they received either an IP injection with vehicle (saline) or hM3Dq-dreadd agonist C21 (2 mg/kg in saline). 60 minutes later the animals were given 2 h fat access. One week later the animals were given two additional fat baseline days on the respective day 3, in randomized design they received the other substance followed by 2 h fat access 60 min after.

## Histology and microscopy
The injection sites of virus, placement of optic fibers and/or cannulas were examined for all experiments and only data from animals with correct injections and, if applicable, fiber or cannula placements were included. After behavioral experiments, animals were transcardially perfused with 4% PFA. Brains were kept in PFA at 4 °C for 24 h before transfer to PBS 0.01% NaAz. Coronal sections of 50, 70 or 100 μm were made on a Vibratome VT1000s (Leica). Images were made with an epifluorescent microscope (Olympus).

**In vivo fiber photometric GCaMP experiments.** To more clearly visualize GCaMP expressing neurons we stained free-floating slices for GFP. After having rinsed the slices with PBS, three times for 10 min

each using Corning® Netwells® inserts (CLS347, sigma Aldrich) (i.e. a rinsing step), sections were incubated with blocking solution (10& NGS, 1% Triton X-100 in PBS) for 1 h on a shaker. After another rinsing step sections were left to incubate overnight at 4 °C in carrier solution (3% NGS, 0.25% Triton x-100 and primary polyclonal anti-GFP antibody (1:1000, GFP-1020, Aves). On the next day, after another rinsing step, sections were incubated in carrier solution containing fluorescence-conjugated secondary antisera raised in goat (Alexa 488, 1:500; Abcam) for 1 h on a shaker in the dark. After another rinsing step sections were mounted with 0.2% gelatine in PBS and left to air dry completely before applying Polyvinyl alcohol mounting medium with DABCO (Merck).

**In vivo optogenetics combined with in vivo fiber photometric dLight experiments.** To more clearly visualize dLight and LHA CoChR-GFP neurons we stained as described above with primary polyclonal anti-GFP antibody (raised in chicken, 1:1000 or 1:2000, GFP-1020, Aves, #GFP87948) and fluorescence-conjugated secondary antisera raised in goat (Alexa 488, 1:500; Abcam).

**In vivo hM3D(Gq)- DREADD experiment.** For better visualization of the rAAV5-hSyn-DIO-hM3D(Gq)-mCherry we utilized primary poly-clonal anti-RFP antibody (raised in rabbit, 1:500, Rockland, 600-401-379, #46317) and fluorescence-conjugated secondary antisera raised in goat (Alexa 488, 1:500; ab150077, Abcam).

**In vivo cannula infusion.** Before anaesthesia, 300 nl red retrobeads (60x diluted in PBS, 100 nl/min; Lumafluor) were infused to visualize the infusion site.

**In vivo optogenetic experiment.** Free floating 75 µm thick sections were stained as described above with primary antibody for TH (mouse anti-TH; 1:2000, Millipore, MAB318, #3782107) and GFP (chicken anti-GFP; 1:2000, GFP-1020, Aves, #GFP87948) to visualize the VTA$_{DA}$ neurons and amplify the GFP signal for histological verification respectively. Secondary antibodies used were anti-chicken 488 (1:500) for GFP and anti-mouse 568 (1:500) for TH.

**Data analysis, statistics and reproducibility**
Data were analyzed with Noldus (Ethovision V9.0), SPSS (IBM V26), Python v3.8.8, MedPC-4 (Med Associates Inc), Igor Pro-8 (Wave-metrics, USA) and Mini Analysis v6.0 (Synaptosoft, USA). Sample size was predetermined on the basis of published studies, experimental pilots and in-house expertise. Animals were randomly assigned to experimental groups. Compiled data are always reported and repre-sented as mean +SEM, with single data points plotted (single cell for electrophysiology and single animal for array tomography, fiber pho-tometric, corticosterone and behavioral experiments). Animals or data points were not excluded from analyses unless noted. When applic-able, statistical tests were One-way ANOVAs, Two-way ANOVAs or Multi-way Repeated-measures ANOVAs. If applicable post hoc con-trasts were performed in case of significant omnibus ANOVAs. If data were not normally distributed statistical testing was performed using the Kolmogorov-Smirnov test. Testing was always performed two-tailed with α = 0.05. Complete outcome of statistical analysis is shown in Supplementary Table 1.

Experiments shown in Figs. 2b, g, 3b, f, 4d, g, 5b, e, and supple-mentary figures 4b, 5c and 6g were repeated independently in at least 4 animals with similar results (exact numbers as indicated in figures legends of respectively Figs. 2c, h, 3c, h, 4e, h, 5c, g and supplementary figures 4e–f, 5e and 6h).

**Reporting summary**
Further information on research design is available in the Nature Portfolio Reporting Summary linked to this article.

## Data availability
The authors declare that all data supporting the findings of this study are available within the article and its supplementary information files. Source Data are provided with this paper. Underlying data of all figures are provided in the Source Data file with this paper and data are fully available from the corresponding author on request. All detailed out-comes of statistical test are provided in Supplementary Table 1. Source data are provided with this paper.

## Code availability
The customized codes used for the analyses in the current study have been made available with the manuscript.

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

## Acknowledgements

We thank M. Mameli, S. Lecca, I. Willuhn and the entire Meye Lab for discussions and critical reading of the manuscript. We thank M. de Wit for help with corticosterone assays. This work was supported by the ERC under the European Union's Horizon 2020 research and innovation programme (grant agreement 804089; ReCoDE), the NWO Veni grant 863.15.012, the Brain & Behavior Research Foundation NARSAD Young

Investigator Grant 25190 and partially by the NWO Gravitation project BRAINSCAPES: A Roadmap from Neurogenetics to Neurobiology (024.004.012). The collaboration between F.J.M. and M.S.-R. was supported by the Young IBRO Regions Connecting Award.

## Author contributions

L.L. and F.M. performed and analyzed electrophysiological recordings and behavioral experiments. B.S. and L.P. helped with behavioral experiments. L.P. and E.S. performed and analyzed fiber photometric recordings, L.B. helped with behavioral analysis. M.S.-R. performed and analyzed array tomography experiments. L.L., I.W.D., M.L. and E.S. performed stereotactic surgeries. F.M. designed the study with L.L. and R.A. L.L. and F.M. wrote the manuscript with the help of all authors.

## Competing interests

The authors declare no competing interests.
