## [Peer Review File · Nature Communications]

Stress-driven potentiation of lateral hypothalamic synapses onto ventral tegmental area dopamine neurons causes increased consumption of palatable foodREVIEWER COMMENTS

Reviewer #1 (Remarks to the Author):

The manuscript by Linders et al., describes the role of LHA-to-VTA glutamatergic inputs on stress induced palatable food consumption. The stress-induced increase in the excitatory drive to VTA DA neurons is somewhat reported, however, this work specifically focused on the importance of LHA-to-VTA inputs. Using patch clamp recording as well as optogenetic manipulation, authors elucidate the synaptic mechanisms underlying stress induced palatable food consumption. The experimental designs are coherent, data are solid, and the manuscript is written well. However, I believe that addressing following concerns will improve the quality of this study.

1. Authors showed that the activating LHA-to-VTA glutamatergic projection selectively increase the release of DA in the mPFC, not medial NAc, of stressed animals. However, detail circuit mechanism has not been addressed. Since DA neurons projecting to mPFC or NAc are distinct neuronal populations, it is likely that synaptic potentiation occurs only on mPFC projecting DA neurons. Authors should show that synaptic potentiation selectively occurs on mPFC-projecting DA neurons, not NAc-projecting DA neurons in stressed animals.

2. Stress-induce specific changes in excitatory synapse by changing AMPA receptor composition (Figure 2). Although Dex treatment induced increase in excitatory synaptic input to DA neurons in ex vivo recording (Figure 5), it is not clear whether Dex can induce same change in AMPA receptor composition. To fully support that the increased synaptic change by stress is through glucocorticoid mediated signaling in VTA, authors should show that Dex can induce the same synaptic types of synaptic changes. Moreover, to further support that GR signaling is important for stress induced increased palatable food consumption, the inhibition of VTA GR signaling should reduce the stress-induced increase in palatable food consumption by stress.

3. Authors showed that fight can acutely increase the activity of VTA-projecting LHA glutamatergic neurons. However, there is no evidence showing that this transient change by fight has any connection to the increased food consumption. To support the role of these neurons in the change in feeding behaviors, authors should perform fiber photometry imaging of these neurons of control and stressed animals during palatable food consumption.

4. Authors showed that the increase in DA release in mPFC when LHA-to-VTA glutamatergic projection is activated in stress animals. This is interesting findings. However, it is not clear the meaning of this increased DA release in stress-induced changes in food consumption. Authors should show that increased DA release has some causal roles for stress-induced palatable food consumption. To address

this, I think it is best for authors to examine DA release in mPFC and NAc of control and stressed animals during normal chow or palatable food consumption.

Reviewer #2 (Remarks to the Author):

In this manuscript, Linders and colleagues carried out a series of functional circuit mapping and electrophysiological recording experiments to show that acute social stress strengthens LHA glutamatergic inputs to VTA DA neurons, which may underlie hedonic overeating in these stressed animals by increasing DA release in the mPFC but not NAc-shell. Although the findings are not completely novel and some of them are even somewhat contradict to previously published work, the results presented still provide some insights into the mechanisms underlying hedonic overeating of palatable food upon psychological and/or emotional stress. While the elaborations of methodological details in the manuscript are applaudable, there are also some obvious concerns for employed techniques and experimental design, which need to be clarified or addressed by additional experiments to firmly support their claimed findings.

Below are my comments for the authors:

1. It has been widely appreciated that behavioral responses to social defeat stress in mice are not homogeneous (Krishnan et al, 2007); some mice show stress susceptibility while others appear to be more resilient. However, the authors did not take this point into their experimental design at all. Indeed, looking at dot plotted bar graphs shown in Figure-1, not every stressed mouse seems to consume a lot more palatable food or sucrose, which raise a question whether those mice used for electrophysiology or fiber photometry recording are all “stress susceptible” or not. It would have been nice if the authors perform some kinds of behavioral screening to distinguish “susceptible” and “resilient” mice first and then test if the strengthening of glutamatergic LHA-VTA circuit is more evident in one group over another.
2. There is an apparent concern of experimental design for FP recording in Figure-2. With viral approach shown here, it is highly likely that GCaMP will also be expressed in extra-LHA glutamatergic neurons and so recorded LHA GCaMP signals could include axon terminal GCaMP originating from these extra-LHA glutamatergic neurons. Additionally, the location of targeted optic cannula shown in representative image (Figure 2b) appears to be the DMH (with high green GCaMP signals in the DMH as well), not the LHA. Were there any mist-targeted cases for this experiment? If so, how was GCaMP signals from those mist-targeted cases look like? Anything difference from what have been observed in the LHA? Lastly, considering potential neuronal toxicity, AAV-retro should have been the better choice than HSV for this experiment.
3. In the fiber photometry recordings for Juvenile mouse interactions, author described that the mice were introduced to a juvenile mouse in a clean cage and after 10 minutes of habituation to the environment and juvenile mouse, photometric signal and video were recorded in synchrony. Similar approach was also used to novel environment test. As animals used in both tests were already

habituated to novel mouse or novel environment, it may not lead to any noticeable changes in photometry signal. It would be better if photometry signal were checked on the first encounter of subject or environment.

4. Although the authors could not see any change in Dlight level at NAc and mPFC using phasic stimulation protocol and later used burst firing protocol to support their circuit mapping, they did not provide any data to support that burst firing of LHA glutamatergic neurons can mimic stress-driven hedonic eating. At the minimum, the authors should attempt artificial strengthening of glutamatergic LHA-VTA synapse by periodic optogenetic stimulation of this circuit to see if it will lead to the preference of palatable food in stress-naïve mice.

5. For Figure-5, the authors show that dexamethasone injection into the VTA increase 2h fat intake, however, they did not confirm this effect is mPFC mediated. It will be very informative if the authors can use parallel Dlight recording experiment to support their circuit mapping shown in Figure-4.

6. Somewhat contradict to the results in this manuscript, another group has previously shown that LHA glutamatergic neurons preferentially synapse onto VTA glutamatergic neurons than DA neurons (Barbano et al, 2020). This discrepancy needs more discussion.

7. "... hypothalamic synapses..." in the title should be changed to "... lateral hypothalamic synapse..."

Reviewer #3 (Remarks to the Author):

Linders et al., perform a series of experiments aimed at understanding the role of lateral hypothalamic projections to the ventral tegmental area in influencing palatable food intake after an acute stressful experience. The authors argue that LHA-Vglut2 neurons projecting to the VTA are responsible for altering excitatory synaptic input to VTA dopamine neurons to increase palatable food consumption after stress. They then argue that this pattern is mediated by action on glucocorticoid receptors within the VTA. Finally, they show that low frequency optogenetic stimulation of LHA-Vglut2-VTA axons can reverse stress-induced changes in synaptic strength and prevent overconsumption of palatable foods. Overall, the manuscript addresses an important question regarding the interplay between stress and food rewards. The behavioral paradigm is powerful, and the general methodology is sound. My primary concerns involve the histological verification of virus and fiber placement as well as the significant limitations that are associated with this type of ex vivo circuit dissection, the latter of which can be primarily addressed through additional discussion. Specific points below.

Major points:

1)The example histology presented throughout the manuscript provides little clarity about the anatomical specificity of recordings/manipulations. In multiple instances, the regions labeled as 'LHA'

appear to be incorrectly drawn to encompass the viral expression. This makes many of the results difficult to interpret. Throughout, only example histology is shown, but these examples often raise more questions than they address. Thus, quantification of the localization of viral expression and/or summaries of injection locations, viral expression, or fiber placements should be included where warranted. Specific examples:

- Fig. 2b: The region labeled 'LHA' encompasses much of the dorsomedial and ventromedial nuclei. This should be corrected. Moreover, the image shows GCaMP largely expressed in the dorsomedial hypothalamus near the ventricle (which has been mostly cropped from the image). This expression pattern makes the claims related to LHA specificity difficult to interpret. The fiber tip is also placed quite lateral in the hypothalamus, with the red outline not accurately overlaid on the visible fiber track. Additional quantification or histological validation demonstrating that recordings were made from LHA should be included.

- Fig. 3b: Similar problem as above. It is quite difficult to see the hypothalamic anatomy, but it appears as if the 'LHA' has been drawn to encompass the DMH and VMH once again. The location of viral expression may be largely outside the LHA proper, but it's difficult to tell from the image. Please include a more informative representative image or some quantification of the expression across the cohort of mice.

- Fig. 4g: the images are too magnified to evaluate the viral expression or location of the fibers. Given the unconvincing histology in earlier figures, an image showing viral expression reasonably restricted to the LHA should be included.

- Fig 5b: This is intended to illustrate the local infusion of Dex and fluorescent beads into the VTA shown in yellow, yet the yellow signal appears widely distributed across the entire field of view (although the most concentrated signal appears to be in the VTA). Is the extra-VTA yellow signal background noise or fluorescent beads? If it is background, I recommend adjusting the levels to make the image more helpful in interpreting the data.

2)The discussion should be expanded to include a significant description of the limitations of the current approach and how that restricts the interpretation of the results. Specifically:

- According to the methods, the ex vivo recordings were performed in the absence of drugs blocking polysynaptic activity. Thus, the monosynaptic nature of the recordings cannot be confirmed. This should be clearly stated throughout the results and mentioned in the discussion.

- Relatedly, the authors have greatly simplified the anatomical complexity of VTA throughout the manuscript. The intra-VTA circuitry consists of multiple cell types each with unique functional sub-populations, projection targets, and inputs. Furthermore, the observation that LHA-Vglut2 neurons have been shown to preferentially target Vglut2+ cells in VTA (Barbano et al., 2020, Neuron) should be discussed in the context of the current results, which assess only Pitx3 and Vgat neurons within the VTA.

- In experiments using hsyn-ChR2, the co-activation of many LHA-VTA cell types is likely occurring. Both GABA and glutamate LHA neurons (as well as other LHA cell types. E.g., orexin, neurotensin receptor) project to VTA and have largely uncharacterized local interactions. This becomes more problematic

when considering the lack of monosynaptic specificity in the recordings. The discussion should directly address the limitations associated with this approach.

- Orexin neurons are known to project to VTA and express Vglut2. A discussion of how the VTA-projecting orexin population fits within the context of the present LHA-Vglut2→VTA results should be included.

3) Although the authors show that GR stimulation increases the AMPA amplitude in DA neurons, the effects of Dex on fat intake (Fig. 5c) are not necessarily driven by direct action on VTA DA neurons. “This suggests that social stress can elevate corticosterone levels, which via local GR-signaling in the VTA can enhance LHAGlu-VTADA synapses and increase palatable fat intake.” This should be qualified as the relationship between GR signaling, DA activity, and palatable food intake is not so clear. It is possible that GR signaling in VTA affects multiple cell types and influences the function of multiple inputs to affect behavior.

4)Does the type of stress influence the observed pattern of results? For example, would you expect similar behavioral and neural outcomes if acute restraint stress or some other non-social stress model was used? A direct test of this is outside the scope of the current manuscript, but a discussion of these questions would be helpful.

Minor points:

5) Intro: please clarify that optogenetic manipulations of LHA-Vglut2 terminals in VTA have been shown to have little effect on feeding (e.g., Nieh et al., 2015, Cell; Nieh et al., 2016, Neuron).

6) Fig. 4d: is the change in GABAR/AMPA ratio caused by a reduction in GABA current or an increase in the AMPA current?

7) Fig 5f: “Putative dopamine neurons” is unjustified given the absence of data showing these cells are dopaminergic (e.g., TH/DAT expression, presence of an Ih current, response to dopaminergic drugs, etc...). Since a high proportion of VTA-GABA cells receive LHA-Glut input, the results and figure legend should be modified to indicate that the recordings are from unidentified VTA neurons or additional data should be included to justify this assertion.

REVIEWER #1 (R1)

“The manuscript by Linders et al., describes the role of LHA-to-VTA glutamatergic inputs on stress induced palatable food consumption. The stress-induced increase in the excitatory drive to VTA DA neurons is somewhat reported, however, this work specifically focused on the importance of LHA-to-VTA inputs. Using patch clamp recording as well as optogenetic manipulation, authors elucidate the synaptic mechanisms underlying stress induced palatable food consumption. The experimental designs are coherent, data are solid, and the manuscript is written well. However, I believe that addressing following concerns will improve the quality of this study.”

1. Effects of stress on specific VTA dopamine projection subpopulations

“Authors showed that the activating LHA-to-VTA glutamatergic projection selectively increase the release of DA in the mPFC, not medial NAc, of stressed animals. However, detail circuit mechanism has not been addressed. Since DA neurons projecting to mPFC or NAc are distinct neuronal populations, it is likely that synaptic potentiation occurs only on mPFC projecting DA neurons. Authors should show that synaptic potentiation selectively occurs on mPFC-projecting DA neurons, not NAc-projecting DA neurons in stressed animals.”

We agree with the reviewer that it is of interest to understand on which subsets of VTA dopaminergic neurons the LHA glutamatergic synapses are affected by stress. In order to address this we have performed the requested experiment in which we identified VTA dopamine neurons based on their projection to either the mPFC or to the NAc medial shell. We observed that social stress leads to synaptic potentiation of LHA glutamatergic inputs to VTA dopamine neurons projecting to the mPFC, but not to the NAc medial shell. These results are in accordance with our *in vivo* dopamine biosensor measures, and they are now described in new Figure 4d-e, and in Supplementary Figure 5a-b.

2. The specific effects of glucocorticoid signaling on LHA-VTA synaptic changes and its contribution to food intake

“Stress-induce specific changes in excitatory synapse by changing AMPA receptor composition (Figure 2). Although Dex treatment induced increase in excitatory synaptic input to DA neurons in ex vivo recording (Figure 5), it is not clear whether Dex can induce same change in AMPA receptor composition. To fully support that the increased synaptic change by stress is through glucocorticoid mediated signaling in VTA, authors should show that Dex can induce the same synaptic types of synaptic changes. Moreover, to further support that GR signaling is important for stress induced increased palatable food consumption, the

inhibition of VTA GR signaling should reduce the stress-induced increase in palatable food consumption by stress.”

We agree with the reviewer that it is interesting to determine whether GR agonist exposure alters the AMPA receptor subunit composition at LHA-VTA synapses in the same way as social stress. We have therefore now experimentally addressed this. We have exposed brain slices to either GR agonist dexamethasone or to its vehicle and subsequently performed patch-clamp experiments from VTA dopamine neurons and optogenetically stimulated LHA inputs. We then assessed the AMPAR rectification index (which we showed to be increased by social stress) and we observed that dexamethasone pre-treatment also increased AMPAR rectification at these synapses, suggesting a switch to GluA2-lacking AMPARs. These data are now shown in new Supplementary Figure 6b-d.

We also agree with the reviewer that it is of interest to address the role of VTA GR signaling in stress-driven food intake. To attempt to address this role we performed multiple experiments. First we gave peripheral administration of GR antagonist mifepristone (or vehicle) to animals during social stress (or control). Unfortunately, this turned out to result in selectively high mortality in the mifepristone-stress group (75%), and we considered it not ethically responsible to continue that experiment. We then tried to locally administer mifepristone in the VTA (during stress or control), but found that it in itself (in absence of stress) already increased food intake, which obscured the stress eating effect, but made it difficult to interpret the extent of residual stress eating. We note that mifepristone unfortunately has affinity for the GR but an even higher affinity for the progesterone receptor (with a known role in food intake), and both receptors are present in the VTA. We therefore unfortunately did not manage to further parse the specific contribution of VTA GR signaling to stress eating. Finally, we would like to note that we performed another experiment (see point 3 below for more details) in which we showed that high frequency stimulation of the LHA_{glu}-VTA pathway is sufficient to induce increased fat intake subsequently (now described in new Suppl. Fig. 6g-i). That experiment suggests that aside from GR agonism also LHA_{glu}-VTA pathway activity (which we show to occur during stress) may contribute to alterations in LHA-VTA synaptic strength and in increased fat intake (i.e. VTA GR antagonism may not be sufficient to fully block the effect). Overall, we hope that the reviewer agrees with us that, while it remains of interest to unravel the specific role of VTA GR in stress eating, this may require approaches that are currently technically not at our disposal, and can provide a future research line.

3. The importance of LHA_{glu}-VTA acute activity during stress for later food intake

“Authors showed that fight can acutely increase the activity of VTA-projecting LHA glutamatergic neurons. However, there is no evidence showing that this transient change by fight has any connection to the increased food consumption. To support the role of these neurons in the change in feeding behaviors, authors should perform fiber photometry imaging of these neurons of control and stressed animals during palatable food consumption.”

We have done two experiments to address the role of stress-driven effects on LHA_{glu}-VTA synapses in relation to food intake.

First, (as requested by Reviewer #2, point 4 below) we determined whether acute bouts of activity of LHA_{glu}-VTA synapses can result in subsequently increased palatable food intake in (otherwise non-stressed) mice. Specifically, we performed an experiment where we *in vivo* stimulated these synapses with bouts of 20 Hz high frequency stimulation. We observed that this in itself was sufficient to drive increased fat (but not chow) intake afterwards. This result is in accordance with a link between the acute activity during social stress in this pathway, and the subsequent altered feeding patterns. These results are now described in new Supplementary Figure 6g-i.

Second, we have also performed the requested fiber photometric measurements from LHA glutamatergic somata (that project to the VTA) during food intake and assessed the effect of stress on this. We show that these neurons are indeed activated during fat consumption. The magnitude of this response did not change after social stress. These findings are in accordance with a role of LHA glutamate neurons in fat binge responses, but also highlight that the relevant stress-driven changes in the pathway may indeed primarily occur at the synapse rather than at the somatic level in the LHA. These results are now described in new Figure 2g-h (and Supplementary Fig. 2f).

4. The role of dopamine release in mPFC in relation to palatable food consumption

“Authors showed that the increase in DA release in mPFC when LHA-to-VTA glutamatergic projection is activated in stress animals. This is interesting findings. However, it is not clear the meaning of this increased DA release in stress-induced changes in food consumption. Authors should show that increased DA release has some causal roles for stress-induced palatable food consumption. To address this, I think it is best for authors to examine DA release in mPFC and NAc of control and stressed animals during normal chow or palatable food consumption.”

We thank the reviewer for these suggestions. We agree that it is interesting to see whether (stress-driven) dopamine responses in mPFC are in principle able to causally contribute to the stress eating behavior. In order to address this in a causal manner as requested, we took advantage of a dopamine 1 receptor Cre mouse line such that we could target mPFC dopamine-sensitive subsets of neurons (i.e. mPFC_{DRD1} neurons). Dopamine can have a prolonged stimulatory effect on this neuronal population (Seamans & Yang, 2004; PMID: 15381316). In order to mimic the effect that stress may exert via dopamine on this population we virally expressed, in a Cre-dependent manner, a stimulatory chemogenetic receptor (hM3Dq) in mPFC_{DRD1} neurons. We observed that stimulation of these neurons with a DREADD agonist was sufficient to drive increased fat binge eating. These data are now described in new Supplementary Figure 5h-i.

REVIEWER #2 (R2)

“In this manuscript, Linders and colleagues carried out a series of functional circuit mapping and electrophysiological recording experiments to show that acute social stress strengthens LHA glutamatergic inputs to VTA DA neurons, which may underlie hedonic overeating in these stressed animals by increasing DA release in the mPFC but not NAc-shell. Although the findings are not completely novel and some of them are even somewhat contradict to previously published work, the results presented still provide some insights into the mechanisms underlying hedonic overeating of palatable food upon psychological and/or emotional stress. While the elaborations of methodological details in the manuscript are applaudable, there are also some obvious concerns for employed techniques and experimental design, which need to be clarified or addressed by additional experiments to firmly support their claimed findings. Below are my comments for the authors:”

1. Stress resilient vs vulnerable phenotypes, and differences for feeding behavior

“It has been widely appreciated that behavioral responses to social defeat stress in mice are not homogeneous (Krishnan et al, 2007); some mice show stress susceptibility while others appear to be more resilient. However, the authors did not take this point into their experimental design at all. Indeed, looking at dot plotted bar graphs shown in Figure-1, not every stressed mouse seems to consume a lot more palatable food or sucrose, which raise a question whether those mice used for electrophysiology or fiber photometry recording are all “stress susceptible” or not. It would have been nice if the authors perform some kinds of behavioral screening to distinguish “susceptible” and “resilient” mice first and then test if the strengthening of glutamatergic LHA-VTA circuit is more evident in one group over another.”

We thank the reviewer for their comment. The reviewer is completely right that in the context of stress-driven depressive-like behaviors resilient vs vulnerable phenotypes can be differentiated. With regards to effects on food intake and body weight, previous studies suggest that the effects of social stress shows inter-individual differences in body weight gain, but not so much in terms of food intake (Bartolomucci et al., 2005; PMID: 15652256 and Bartolomucci et al., 2009; PMID: 19180229). The reviewer correctly points out that there is variability in our Figure 1 subpanels when considering individual post-stress days and individual nutrients consumed. This in part emerges because some stressed mice particularly eat more fat but do not drink more sucrose (or vice versa) on a given post-stress day (often compensated on the next day). If instead we collapse the total palatable food intake from fat and sugar (from Fig. 1b-c) over the two post-stress days, we see a homogenous effect of stress without indication of sub-clustering (indeed similar SEM in the control group and stress group). We present these data here for the reviewer (Figure X below).

Figure X. Average palatable food intake (fat and sucrose) collapsed over two post-stress days. SEM for control group is 0.98 kCal and 0.84 kCal for the stress group, suggesting relative homogeneity in the stress response for short-term food intake.

We agree with the reviewer that it is interesting to see which neural circuit changes go along with inter-individual differences in eventual stress-driven body weight gain. However, we hope the reviewer agrees with us that, considering: (a) relative food intake homogeneity of the stress effect, and (b) the possibility that adding resilience/vulnerability tests may influence the plasticity and behavior, we think this is something that is better addressed in a dedicated future study.

2. Specificity of Fiber photometric recordings from LHA glutamate neurons projecting to VTA

“There is an apparent concern of experimental design for FP recording in Figure-2. With viral approach shown here, it is highly likely that GCaMP will also be expressed in extra-LHA glutamatergic neurons and so recorded LHA GCaMP signals could include axon terminal GCaMP originating from these extra-LHA glutamatergic neurons. Additionally, the location of targeted optic cannula shown in representative image (Figure 2b) appears to be the DMH (with high green GCaMP signals in the DMH as well), not the LHA. Were there any mist-targeted cases for this experiment? If so, how was GCaMP signals from those mist-targeted cases look like? Anything difference from what have been observed in the LHA? Lastly, considering potential neuronal toxicity, AAV-retro should have been the better choice than HSV for this experiment.”

We thank the reviewer for their observations. We address the points below:

2a. With regards to the viral strategy leading to potential recordings being made from non-LHA glutamatergic neurons. In our strategy we ensured that we included only mice in which under the fiber there were clear cell bodies in LHA. Aside from cell bodies, there are also dendritic/axonal processes present. Those likely mainly belong to LHA glutamate neurons, but we do agree with the reviewer that it is possible that a subset could represent some non-LHA neurons whose processes extend into the LHA. In order to ensure that LHA glu cell

bodies themselves are stress-sensitive, we have now performed fiber photometric recordings with AAV-DIO-GCaMP expressed directly in the LHA of Vglut2-cre mice. In this way, though it does not only tap into VTA-projectors, it is unlikely that there are *en passant* fibers or extending dendrites from extra-LHA sites running under the optic fiber. With this corroborating experiment we similarly observe a strong increase in activity during fights. Together with our retrogradely expressed GCaMP data, this makes it likely that indeed LHA_{Glu} neurons projecting to VTA are activated by social stress. These data are now shown in new Supplementary Fig. 2g.

2b. With regards to the location of the optic fibers. We have now included an overview of all the optic fiber end points in a schematic overview. From this emerges that our recording tips occur at various sites in the LHA, most of them quite lateral. The previous example image, for which we agree that the localization was at the border of LHA and other hypothalamic sites, was not entirely representative and we have put a more representative image instead. As requested by the reviewer we have also now included fiber photometric recording examples of discarded recordings, which show that these neither produced fight reactivity nor fat intake reactivity (a new experiment we ran, see point 4 for R1). These analyses are now shown in new Supplementary Fig. 2f.

2c. We agree that toxicity is a factor always of concern with viral approaches, with different trade-offs of toxicity/tropism occurring between different types of vectors. The HSV-LS1L-GCaMP vector has previously been successfully used in studies also probing the glutamatergic LHA-VTA pathway (e.g. Barbano et al., 2020, where viral incubation times of over 8 weeks still yielded successful recordings). Moreover, we have now also shown with an AAV-DIO-GCaMP (point 2a) that LHA glu neurons react to fights. So overall, while we acknowledge the point that retro-AAV may have certain benefits, we would argue that our dataset shows that LHA_{glu} neurons projecting to the VTA show reactivity to (social) stress.

3. Absence of novelty effects in fiber photometric recordings from LHA_{glu}-VTA pathway

“In the fiber photometry recordings for Juvenile mouse interactions, author described that the mice were introduced to a juvenile mouse in a clean cage and after 10 minutes of habituation to the environment and juvenile mouse, photometric signal and video were recorded in synchrony. Similar approach was also used to novel environment test. As animals used in both tests were already habituated to novel mouse or novel environment, it may not lead to any noticeable changes in photometry signal. It would be better if photometry signal were checked on the first encounter of subject or environment.”

We thank the reviewer for their suggestion. The primary goal of our usage of the juvenile mouse and the novel environment experiments was to parse out non-anxiogenic types of social interaction (in case of the juvenile) and general bouts of (non-anxiogenic) physical

activity in case of the open field. For this reason we indeed ensured that any neophobia of those experiences was relatively diminished via prior habituation. However, we do agree with the reviewer that it is interesting to assess to which extent the LHA_{glu}-VTA pathway is sensitive to relative novelty. To that end we compared first vs last juvenile social interactions, but did not observe a significant difference in the magnitude of these responses. These data are now shown in Suppl Fig. 2d.

4. Mimicking stress eating responses with high frequency stimulation of LHA_{glu}-VTA

“Although the authors could not see any change in Dlight level at NAc and mPFC using phasic stimulation protocol and later used burst firing protocol to support their circuit mapping, they did not provide any data to support that burst firing of LHA glutamatergic neurons can mimic stress-driven hedonic eating. At the minimum, the authors should attempt artificial strengthening of glutamatergic LHA-VTA synapse by periodic optogenetic stimulation of this circuit to see if it will leads to the preference of palatable food in stress-naïve mice.”

We agree with the reviewer that it is very interesting to see if the social fight-driven activity bursts in LHA_{glu} neurons projecting to the VTA are sufficient to drive subsequent palatable food intake. To address this we have now indeed performed an experiment in which we applied bouts of high frequency (20 Hz) optogenetic stimulation of LHA_{glu}-VTA synapses in non-stressed mice. We observe that subsequent to this stimulation, mice indeed consume more fat (but not chow). These data are now shown in Supplementary Fig. 6f-i.

5. The potential role of the mPFC in mediating stress-driven food intake effects

“For Figure-5, the authors show that dexamethasone injection into the VTA increase 2h fat intake, however, they did not confirm this effect is mPFC mediated. It will be very informative if the authors can use parallel Dlight recording experiment to support their circuit mapping shown in Figure-4.”

We thank the reviewer for their comment. The reviewer asked for evidence that mPFC partly mediates effects of stressors on food intake. We agree that it is of interest to understand whether the effects of stressors on mPFC may be relevant in the context of stress eating behaviors. This indeed also touches base with Reviewer 1’s point #4 (see above). To address this we have opted to assess whether dopamine-sensitive neuronal populations in the mPFC can contribute to binge eating. In order to address this in a causal manner we opted to mimic the stimulatory effects that (stress-driven) dopamine signaling in mPFC can have on its dopamine 1 receptor (DRD1) neuronal population (Seamans & Yang, 2004; PMID: 15381316). For this we used a DrD1-Cre line in which we virally expressed, in a Cre-dependent manner, a stimulatory chemogenetic receptor (hM3Dq) in mPFC_{D1R} neurons. We observed that chemogenetic activation of these neurons was sufficient to drive increased binge eating of fat. These data are now described in new Supplementary Figure

5h-i. These data are in accordance with a potential role of stress-driven mPFC dopamine signaling in relation to subsequent food choices.

6. Role of LHA projections to VTA glutamatergic neurons?

“Somewhat contradict to the results in this manuscript, another group has previously shown that LHA glutamatergic neurons preferentially synapse onto VTA glutamatergic neurons than DA neurons (Barbano et al, 2020). This discrepancy needs more discussion.”

The reviewer is indeed correct that aside from LHA connectivity with VTA_{DA} and VTA_{GABA} neurons, there is also substantial LHA connectivity with VTA_{glu} neurons. We do not think that our findings are in contradiction with these results (as connectivity with all these different cell types is found across studies). We agree that VTA glutamatergic neurons also represent a very interesting substrate through which effects of stress on LHA-VTA circuitry may manifest. As requested by the reviewer, we have now touched on this point in the Discussion as an important avenue for further research.

7. Title of the manuscript

“... hypothalamic synapses...” in the title should be changed to “... lateral hypothalamic synapse...”.

We agree with this suggestion and we have changed the title accordingly.

REVIEWER #3 (R3)

“Linders et al., perform a series of experiments aimed at understanding the role of lateral hypothalamic projections to the ventral tegmental area in influencing palatable food intake after an acute stressful experience. The authors argue that LHA-Vglut2 neurons projecting to the VTA are responsible for altering excitatory synaptic input to VTA dopamine neurons to increase palatable food consumption after stress. They then argue that this pattern is mediated by action on glucocorticoid receptors within the VTA. Finally, they show that low frequency optogenetic stimulation of LHA-Vglut2→VTA axons can reverse stress-induced changes in synaptic strength and prevent overconsumption of palatable foods. Overall, the manuscript addresses an important question regarding the interplay between stress and food rewards. The behavioral paradigm is powerful, and the general methodology is sound. My primary concerns involve the histological verification of virus and fiber placement as well as the significant limitations that are associated with this type of ex vivo circuit dissection, the latter of which can be primarily addressed through additional discussion. Specific points below.

Major points:”

1. More clarity on histological representations

“The example histology presented throughout the manuscript provides little clarity about the anatomical specificity of recordings/manipulations. In multiple instances, the regions labeled as ‘LHA’ appear to be incorrectly drawn to encompass the viral expression. This makes many of the results difficult to interpret. Throughout, only example histology is shown, but these examples often raise more questions than they address. Thus, quantification of the localization of viral expression and/or summaries of injection locations, viral expression, or fiber placements should be included where warranted.”

The reviewer is correct and we apologize for where there was a lack of clarity. As requested we have now shown more representative examples and corrected, where applicable, the delineation of the LHA. See further specifics below.

“Specific examples:“

“Fig. 2b: The region labeled ‘LHA’ encompasses much of the dorsomedial and ventromedial nuclei. This should be corrected. Moreover, the image shows GCaMP largely expressed in the dorsomedial hypothalamus near the ventricle (which has been mostly cropped from the image). This expression pattern makes the claims related to LHA specificity difficult to interpret. The fiber tip is also placed quite lateral in the hypothalamus, with the red outline not accurately overlaid on the visible fiber track. Additional quantification or histological validation demonstrating that recordings were made from LHA should be included.”

With regards to the fiber photometric data in Figure 2, we have added (also in accordance with Reviewer 2's point #2) the quantification of fiber tips and have shown a more representative image in light of that quantification.

"Fig. 3b: Similar problem as above. It is quite difficult to see the hypothalamic anatomy, but it appears as if the 'LHA' has been drawn to encompass the DMH and VMH once again. The location of viral expression may be largely outside the LHA proper, but it's difficult to tell from the image. Please include a more informative representative image or some quantification of the expression across the cohort of mice."

Done.

"Fig. 4g: the images are too magnified to evaluate the viral expression or location of the fibers. Given the unconvincing histology in earlier figures, an image showing viral expression reasonably restricted to the LHA should be included."

Agreed, we have now done so.

"Fig 5b: This is intended to illustrate the local infusion of Dex and fluorescent beads into the VTA shown in yellow, yet the yellow signal appears widely distributed across the entire field of view (although the most concentrated signal appears to be in the VTA). Is the extra-VTA yellow signal background noise or fluorescent beads? If it is background, I recommend adjusting the levels to make the image more helpful in interpreting the data."

We thank the reviewer for the suggestion. They are right that the yellow pseudocoloring of the fluorescence was not ideal in terms of contrast. We have therefore reverted back to the original red fluorescence of the co-injected beads (with the Dex) which indeed makes it easier to see the specificity of the injection in the VTA.

2. Expanding the discussion to include more limitations on the ex vivo circuit mapping approaches

"The discussion should be expanded to include a significant description of the limitations of the current approach and how that restricts the interpretation of the results. Specifically:

2a. According to the methods, the ex vivo recordings were performed in the absence of drugs blocking polysynaptic activity. Thus, the monosynaptic nature of the recordings cannot be confirmed. This should be clearly stated throughout the results and mentioned in the discussion."

We have added to both the Results section and to the Discussion that the experiments were performed in absence of TTX as a polysynaptic blocker, and that this means that the monosynaptic nature cannot be entirely confirmed.

2b. “Relatedly, the authors have greatly simplified the anatomical complexity of VTA throughout the manuscript. The intra-VTA circuitry consists of multiple cell types each with unique functional sub-populations, projection targets, and inputs. Furthermore, the observation that LHA-Vglut2 neurons have been shown to preferentially target Vglut2+ cells in VTA (Barbano et al., 2020, Neuron) should be discussed in the context of the current results, which assess only Pitx3 and Vgat neurons within the VTA.”

We agree with the reviewer that VTA anatomical complexity is substantial and was perhaps not entirely done justice in the previous rendition. Similarly the connection between LHA_{glu} and VTA_{glu} neurons needed to be highlighted further. We have corrected both these matters in the Discussion section (also see Reviewer 2’s, point 6).

2c. “In experiments using hsyn-ChR2, the co-activation of many LHA-VTA cell types is likely occurring. Both GABA and glutamate LHA neurons (as well as other LHA cell types. E.g., orexin, neurotensin receptor) project to VTA and have largely uncharacterized local interactions. This becomes more problematic when considering the lack of monosynaptic specificity in the recordings. The discussion should directly address the limitations associated with this approach.

2d. Orexin neurons are known to project to VTA and express Vglut2. A discussion of how the VTA-projecting orexin population fits within the context of the present LHA-Vglut2∠VTA results should be included.”

We agree on both counts. We have addressed this further in the Discussion section, where we explain that in our study we have generally co-activated multiple LHA-VTA pathways, and that we have not pinpointed which of the multiple glutamatergic subpathways (e.g. orexin positive or negative) from LHA to VTA are affected.

3. Locality of action of GR agonist in or near VTA to induce potentiation of LHA_{glu}-VTA synapses

“Although the authors show that GR stimulation increases the AMPA amplitude in DA neurons, the effects of Dex on fat intake (Fig. 5c) are not necessarily driven by direct action on VTA DA neurons. “This suggests that social stress can elevate corticosterone levels, which via local GR-signaling in the VTA can enhance LHAGlu-VTADA synapses and increase palatable fat intake.” This should be qualified as the relationship between GR signaling, DA activity, and palatable food intake is not so clear. It is possible that GR signaling in VTA affects multiple cell types and influences the function of multiple inputs to affect behavior.”

We agree with the reviewer. The effects of the GR agonist within a slice suggests some locality of action in or near VTA and LHA terminals, but we fully agree that it can remain a complex local interplay through which this occurs. We have now further nuanced this segment.

4. Reflections on whether different types of stressors may have similar neural and behavioral outcomes

“Does the type of stress influence the observed pattern of results? For example, would you expect similar behavioral and neural outcomes if acute restraint stress or some other non-social stress model was used? A direct test of this is outside the scope of the current manuscript, but a discussion of these questions would be helpful.”

We agree that this remains a very interesting question. Although (moderate) social stress is one of the stressors in literature found to most consistently increase reward seeking behavior, it is not the only one to do so. Indeed, restraint stress is another example of an experience able to drive increased intake of palatable food (Pecoraro et al., 2004; PMID: 15142987). We have now added this to the Discussion and stated that it remains an interesting question to which extent such different stressors converge on the same neural circuit changes that we observe here with social stress.

“Minor points:”

5. *“Intro: please clarify that optogenetic manipulations of LHA-Vglut2 terminals in VTA have been shown to have little effect on feeding (e.g., Nieh et al., 2015, Cell; Nieh et al., 2016, Neuron).”*

Done.

6. *“Fig. 4d: is the change in GABAR/AMPA ratio caused by a reduction in GABA current or an increase in the AMPA current?”*

In view of the alterations in (non-input specific) sEPSCs, but not sIPSCs we consider it likely that the balance change mainly reflects alterations in the AMPAR currents (and the LHA input-specific investigations on AMPAR/NMDAR, AMPAR rectification index and array tomographic data are in agreement with this). Nevertheless, despite evidence that LHA_{glu} synapses onto VTA_{DA} cells get potentiated by stress, we do not entirely rule out that LHA_{GABA} synapses onto VTA_{DA} cells may also exhibit some adaptations (on other metrics than the evaluated and unaltered paired pulse ratios in Fig. 3e and Suppl. Fig 4f).

7. *“Fig 5f: “Putative dopamine neurons” is unjustified given the absence of data showing these cells are dopaminergic (e.g., TH/DAT expression, presence of an I_h current, response to*

dopaminergic drugs, etc...). Since a high proportion of VTA-GABA cells receive LHA-Glut input, the results and figure legend should be modified to indicate that the recordings are from unidentified VTA neurons or additional data should be included to justify this assertion.”

Agree. We have now corrected this.

Overall we would again like to thank the reviewers for their comments and suggestions, which we feel have greatly helped further strengthen the conclusions of our manuscript.

Kind regards,

Frank Meye

REVIEWERS' COMMENTS

Reviewer #1 (Remarks to the Author):

The authors have made considerable revisions in response to my initial review. They have added results from a number of additional experiments and also modified language to account for limitations.

Authors have done a very thorough and careful job addressing concerns of other reviewers' too.

Reviewer #2 (Remarks to the Author):

The authors performed substantial amounts of additional experiments and addressed most of my major concerns. Although I still think their argument on homogenous effect of stress on palatable food intake is somewhat far-fetched, I think the overall quality of manuscript is now significantly improved. I don't have any further major concerns.

Reviewer #3 (Remarks to the Author):

I commend the authors for a thorough revision that has improved the impact of the manuscript substantially. All my concerns have been addressed.

Dear Editor,

We thank the reviewers for unanimously acknowledging that we have made considerable revisions to the benefit of the study. We are happy that the reviewers have no residual concerns after these revisions. We cite the reviewer comments verbatim below for convenience:

Reviewer #1 (Remarks to the Author):

“The authors have made considerable revisions in response to my initial review. They have added results from a number of additional experiments and also modified language to account for limitations. Authors have done a very thorough and careful job addressing concerns of other reviewers' too.”

Reviewer #2 (Remarks to the Author):

“The authors performed substantial amounts of additional experiments and addressed most of my major concerns. Although I still think their argument on homogenous effect of stress on palatable food intake is somewhat far-fetched, I think the overall quality of manuscript is now significantly improved. I don't have any further major concerns.”

Reviewer #3 (Remarks to the Author):

“I commend the authors for a thorough revision that has improved the impact of the manuscript substantially. All my concerns have been addressed.”

We thank the reviewers for their very constructive role in this process.

As requested in the author checklist, we also add here for convenience the max 250 character summary of our work for the website/e-alerts:

Stress can increase the consumption of rewarding food, which contributes to obesity and binge eating disorders. Here the authors show that stress eating depends on a strengthened connection between the lateral hypothalamus and the dopamine system.

Kind regards,

Frank Meyer